

# Climatological distribution of dissolved inorganic nutrients in the Western Mediterranean Sea (1981-2017)

**Malek Belgacem [1,2], Katrin Schroeder [1], Alexander Barth [3], Charles Troupin [3], Bruno Pavoni [2], Jacopo Chiggiato [1]**

[1]CNR-ISMAR, Arsenale Tesa 104, Castello 2737/F, 30122 Venezia, Italy

[2]Dipartimento di Scienze Ambientali Informatica e Statistica, DAIS, Università Ca' Foscari Venezia, Campus Scientifico Mestre, Italy

[3]GeoHydrodynamics and Environment Research, GHER, University of Liège, Quartier Agora, Allée du 6-Août, 17, Sart Tilman, 4000 Liège 1, Belgium

Correspondence: Malek Belgacem (malek.belgacem@ve.ismar.cnr.it)

**Abstract**

The Western MEDiterranean Sea BioGeochemical Climatology (BGC-WMED) presented here is a product derived from in situ observations. Annual mean gridded nutrient fields for the period 1981-2017, and its sub-periods 1981-2004 and 2005-2017, on a horizontal 1/4° × 1/4° grid have been produced. The biogeochemical climatology is built on 19 depth levels and for the dissolved inorganic nutrients nitrate, phosphate and orthosilicate. To generate smooth and homogeneous interpolated fields, the method of the Variational Inverse Model (VIM) was applied. A sensitivity analysis was carried out to assess the comparability of the data product with the observational data. The BGC-WMED has then been compared to other available data products, i.e. the medBFM biogeochemical reanalysis of the Mediterranean Sea and the World Ocean Atlas18 (WOA18) (its biogeochemical part). The BGC-WMED product supports the understanding of inorganic nutrient variability in the western Mediterranean Sea, in space and in time, but can also be used to validate numerical simulations making it a reference data product.

**Keywords:** western Mediterranean Sea, climatology, inorganic nutrients, in situ observations.

## 1 Introduction

Ocean life relies on the loads of marine macro-nutrients (nitrate, phosphate and orthosilicate) and other micro-nutrients within the euphotic layer. They fuel phytoplankton growth, maintaining thus the equilibrium of the food web. These nutrients may reach deeper levels through vertical mixing/upwelling,



and remineralization of sinking organic matter. Ocean circulation and physical processes continually
drive the large-scale distribution of chemicals (Williams and Follows, 2003) toward a homogeneous
distribution. Therefore, nutrient dynamics is important to understand the overall ecosystem productivity
and carbon cycles.  In general, the surface layer is depleted in nutrients in low latitude regions (Sarmiento
and Toggweiler, 1984), but in some ocean regions, called high nutrient low chlorophyll (HNLC) regions,
nutrient concentrations tend to be anomalously high, particularly in areas of the North Atlantic and
Southern Ocean, as well as in the eastern equatorial Pacific, and in the North Pacific; see e.g. Pondaven
et al. (1999). In the Mediterranean, the surface layer is usually nutrient-depleted. Most studies show that
nitrate is the most common limiting factor for primary production in the global ocean (Moore et al.,
2013), while others evidence that phosphate may be a limiting factor in some specific areas, as is the
case of the Mediterranean Sea (Diaz et al., 2001; Krom et al., 2004).
Being an enclosed marginal sea, the Mediterranean Sea exhibits an anti-estuarine circulation,
responsible for its oligotrophic character (Bethoux et al., 1992; Krom et al., 2010) and acting like a
subtropical anticyclonic gyre. The Atlantic Water (AW), characterized by low-salinity and low-nutrient
content, enters the Western Mediterranean Sea (WMED) at the surface, through the Strait of Gibraltar,
and moves toward the Eastern Mediterranean Sea (EMED), crossing the Sicily Channel (Fig. 1). In the
Levantine and in the Cretan Sea, the AW becomes saltier, warmer and denser, and it sinks to
intermediate levels (200-500 m) to form the Intermediate Water (IW, Schroeder et al., 2017). The IW
(which may be further called Levantine or Cretan Intermediate Water, LIW or CIW) flows westward
across the entire Mediterranean Sea to the Atlantic Ocean (Fig. 1).  As for the deep layer, the Western
Mediterranean Deep Water (WMDW or DW) is formed in the Gulf of Lion through deep convection
(Testor et al., 2018) while the Eastern Mediterranean Deep Water (EMDW) is formed in the Adriatic
Sea and occasionally in the Aegean Sea (Lascaratos et al., 1999; Roether et al., 1996, 2007).


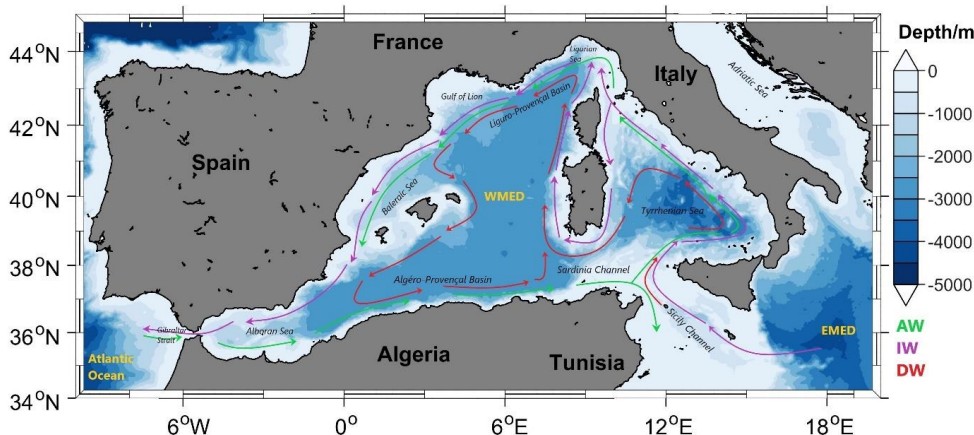

**Figure 1.** Map of the western Mediterranean Sea showing the main regions with a sketch of the AW, IW and DW major paths.

The Mediterranean Sea is known to be a hotspot for climate change (Giorgi, 2006). During the early 1990s, the Deep Water (DW) formation area of the EMED shifted from the Adriatic Sea to the Aegean Sea. This event is known as the Eastern Mediterranean Transient (EMT; Roether et al., 1996, 2007, 2014; Roether and Schlitzer, 1991; Theocharis et al., 2002). As a consequence, the intermediate and deep waters of the EMED became saltier and warmer (Lascaratos et al., 1999; Malanotte-Rizzoli et al., 1999). The EMT affected the WMED as well, not only changing the thermohaline characteristics of the IW and concurring to the preconditioning of the Western Mediterranean Transition (WMT; Schroeder et al., 2016), which set the beginning of a rapid warming and salting of the deep layers in the WMED since 2005 (Schroeder et al., 2006; Schroeder et al., 2010, 2016; Piñeiro at al., 2019). Over the last decade, it has been evidenced that heat and salt content have been increasing in all over the deep western basin (Schroeder et al., 2016).

Changes in circulation due to an increased stratification limit the exchange of materials between the nutrient-rich deep layers and the surface layers. Understanding the peculiar oligotrophy of the Mediterranean Sea is still a challenge, since there is not an exact quantification of nutrient sinks and sources. Studies like Crispi et al. (2001), Ribera d'Alcalà (2003), Krom et al. (2010) and Lazzari et al. (2012) related the horizontal spatial patterns in nutrient concentrations mainly to the anti-estuarine circulation which exports nutrients to the Atlantic Ocean, showing a decreasing tendency of nutrient concentrations toward east, as opposed to the salinity horizontal gradient. These variations, together with the anthropogenic perturbations affect the spatial distribution of nutrients (Moon et al., 2016) while temporal variability is still unresolved.

De Fommervault et al. (2015) reported a decreasing phosphate and an increasing nitrate concentrations trend between 1990 and 2010, based on a time series (DYFAMED) in the Ligurian Sea, while Moon et



al. (2016) evidenced an increase between 1990 and 2005 and a gradual decline after 2005 in both nitrate
and phosphate in the WMED and EMED.
At the global scale, most of the biogeochemical descriptions are based on model simulations and satellite
observations (using sea surface chlorophyll concentrations (Salgado-Hernanz et al., 2019) but also on
the increasing use of Biogeochemical Argo floats (D'Ortenzio et al., 2020; Lavigne, 2015; Testor et al.,
2018), since in situ observations of nutrients are generally infrequent and scattered in space and time.
For this reason, climatological mapping is often applied to sparse in situ data in order to understand the
biogeochemical state of the ocean representing monthly, seasonally, or annually averaged fields.
Levitus (1982) was the first to generate objectively analyzed fields of potential temperature, salinity,
and dissolved oxygen, and to produce a climatological atlas of the world ocean.
Later on the World Ocean Atlas (WOA), the North Sea climatologies and the Global ocean Carbon
Climatology resulting from GLODAP data product (Key et al., 2004) used the Cressman analysis (1956)
with modified Barnes scheme (Barnes 1964, 1994). In 1994, the first World Ocean Atlas (WOA94;
Conkright et al., 1994) was released integrating temperature, salinity, oxygen, phosphate, nitrate, and
silicate observations. Every four years there is a renewed release of the WOA with an updated World
Ocean database (WOD).
On the regional scale, the first salinity and temperature climatology of the Mediterranean Sea was
produced by Hecht et al. (1988) for the Levantine Basin. Picco (1990) was also among the first to
describe the WMED between 1909 and 1987. In 2002, the Medar/Medatlas group (Fichaut et al., 2003)
archived a large amount of biogeochemical and hydrographic in situ observations for the entire region
and used the Variational Inverse Model (VIM; Brasseur, 1991) to build seasonal and interannual gridded
fields. In 2006, the SeadataNet EU project integrated all existing data, to provide temperature and
salinity regional climatology products for the Mediterranean Sea using VIM as well (Simoncelli et al.,
2016), and dissolved inorganic nutrients (nitrate, phosphate and silicate) 6-years centered average from
1965 to 2017 are available on the EMODnet chemistry portal (https://www.emodnet-chemistry.eu/).
Within this context, in this study regional climatological fields of in situ nitrate, phosphate and silicate,
using the Data Interpolation Variational Analysis (DIVAnd; Barth et al., 2014) are presented here,
providing a high-resolution field contributing to the existing products (Table 1).
The aim of this study is to give a synthetic view of the biogeochemical state of the WMED, to evaluate
the mean state of inorganic nutrients over 36 years of in situ observations and to investigate upon a
biogeochemical signature of the effect of the WMT .
The paper is organized as follows, section 2 describes the data sources used and the quality check;
section 3 is devoted to the methodology, section 4 presents the main results including a comparison of
the new climatology with other products. At the end, we address the change in biogeochemical
characteristics before and after WMT.



**Table 1.** Overview of the existing inorganic nutrient climatologies in the Western Mediterranean Sea.

| Climatology | WOA | EMODnet | BGC-WMED (Present study) |
|---|---|---|---|
| Reference | (Garcia et al., 2019) | (Míguez et al., 2019) | (Belgacem et al., 2021) |
| Year of release | 2018 | 2018 | 2021 |
| Parameter | Nitrate/ Phosphate/ Silicate | Nitrate/ Phosphate/ Silicate | Nitrate/ Phosphate/ Silicate |
| Vertical resolution | Seasonal: 43 levels 0-800m Annual: 102 levels 0-5500m | 21 standard depth 0-1100m (nitrate) 0-1500m (phosphate) 0-1500m (silicate) | 19 levels 0-1500m |
| Horizontal resolution | 1° latitude longitude grid | 1/8° | 1/4° |
| Observation time span | 1955-2017 | 1970 to 2016 (nitrate) 1960 to 2016 (phosphate) 1965 to 2016 (silicate) | 1981-2017 |
| Area | Global | Mediterranean Sea | Western Mediterranean Sea |
| Temporal resolution | Season Decadal | Season 6 year running averages | whole observational period, and two sub-intervals (1981-2004, 2005-2017) |
| Climatology analysis method/ parameter | Objective analysis | DIVA (Data-Interpolating Variational Analysis) tool | DIVAnd (Data-Interpolating Variational Analysis N-dimension) |
| Correlation length | - | optimized and filtered vertically and a seasonally averaged profile was used. | optimized and filtered vertically and horizontally |
| Signal to noise ratio | - | A constant value = 1 | A constant value = 0.5 |
| Background field | - | the data mean value is subtracted from the data. | the data mean value is subtracted from the data |
| Detrending | - | No | No |
| Advection constraint applied | - | No | No |

## 2 Data

The climatological analysis depends on the temporal and spatial distribution of the available in situ data, and the reliability of these observations. Due to the scarcity of biogeochemical observations in the WMED, merging and compiling data from different sources was necessary.

### 2.1 Data Sources

In total, 2253 in situ inorganic nutrient profiles are the base of the biogeochemical climatology of the WMED (Table 2) that is described here. These profiles cover the period 1981-2017 and come from four main sources, i.e. the Medar/MEDATLAS (1981-1996, Fichaut et al., 2003), the recently published CNR_DIN_WMED_20042017 biogeochemical dataset (2004-2017) (Belgacem et al., 2020), the SeaDataNet data product (2001-2016) and other data collected during MedSHIP programs (Schroeder et al., 2015), GLODAPv2 (https://www.glodap.info/) and CARIMED (http://hdl.handle.net/10508/11313) data products. All datasets are a selection of oceanographic cruises carried out within the framework of European projects or by regional institutions. Data were chosen to ensure high spatial coverage (Fig. 3).

**Table 2.** Number of inorganic nutrient profiles and data sources.

| Source | N. of profiles | N. of observations | Link |
|---|---|---|---|
| MEDATLAS | 940 | 8839 | https://odv.awi.de/data/ocean/medatlasii/ |
| SEADATANET | 523 | 15388 | http://seadatanet.maris2.nl/v_rsm/content.asp?screen=0&history=yes |
| CNR_DIN_WMED_20042017 | 737 | 8324 | https://doi.org/10.1594/PANGAEA.904172 |
| Other cruises | 53 | 515 | Medship programs; GLODAPv2; CARIMED (not yet available online, personal communication by Marta Álvarez) |
| ∑ | *2253* | *33066* | - |

## 2.2 Data distribution

The data distribution per year is shown in Figure 2a. Most observations were collected between 1981 and 1995, and between 2004 and 2017, with a marked gap between 1997 and 2003. Measurement distribution differs from month to month (Fig.2b) and tends to be biased towards the warm season. Very few measurements have been made during December-January-February, while June and July are the months with the highest number of available observations (>7000). Consequently, the climatological product may be considered as being more representative of spring and summer conditions.

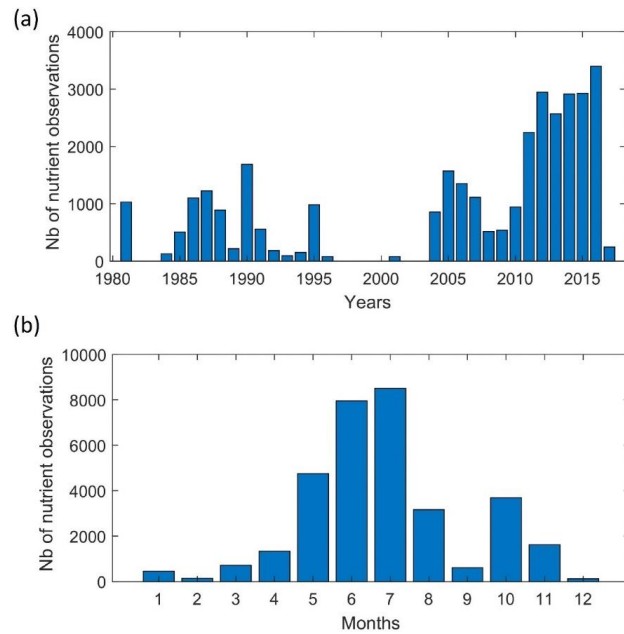

**Figure 2**. Temporal distribution of nutrient observations used for producing the BGC-WMED fields (1981-2017), (a) yearly distribution and (b) monthly distribution.

Fig. 3a shows the regional distribution of nutrient measurements, while Fig. 3b indicates the number of
observations found in each depth range around the standard levels chosen for the vertical resolution of
the climatology.
Hydrological and biogeochemical measurements have always been repeatedly collected along several
repeated transects, known as key regions as the Sicily Channel and the Algéro-Provençal subbasin;
likewise, the northern WMED is a well sampled area, as it is an area of DW formation. Observation
density is still scarce (less than 100 observations) in some areas like the northern Tyrrhenian Sea.
The total number of measurements at each depth range underlines similar remarks, an uneven
distribution that needs to be considered in the selection of the vertical resolution to estimate the
climatological fields. Though, the use of 36 years of nutrient measurements to generate the
climatological fields significantly reduces the error field. In our case and taking into account the irregular
distribution in seasons and different years. A climatological gridded field was computed by analyzing
observations of three time periods regardless of the month: 1981-2017 and the subsets 1981-2004 and
2005-2017. We chose these subsets to investigate the effect of the WMT on nutrient distribution.

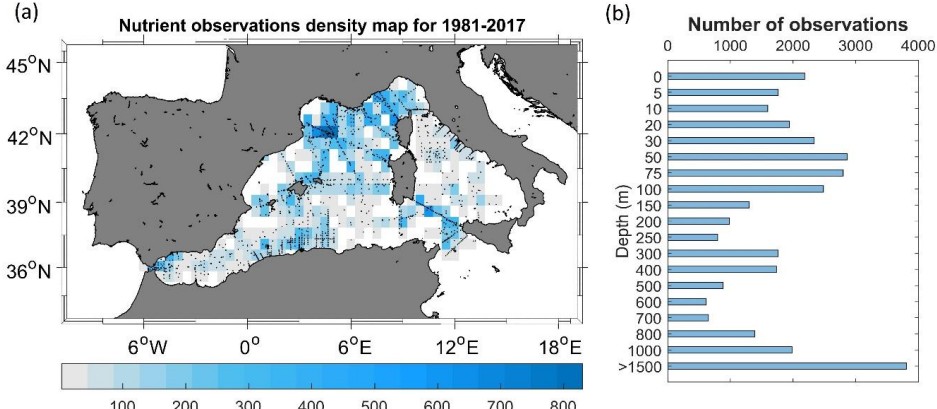


**Figure 3.** (a) Nutrient data density used for climatology analysis. Observations are binned in a regular
$1/2° \times 1/2°$ latitude, longitude grid for each year over the period 1981-2017. Location of the stations
included in the analysis are shown as black dots; (b) data distribution per depth range (i.e. at 800 m,
observations between 800-1000 m are included).
2.3 Data quality check
Data were gathered from different data sources, thus before merging them, observations were first
checked for duplicate (the number of profiles listed in Table 2 refers to all data after removing duplicate
measurements). The criteria to detect and remove duplicate is simple: observations collected during
same cruises extracted from the different sources were removed. Since profiles were measured during





specific cruise (identified with a unique identification code) at specific time. Data from duplicate cruise
are removed.
Then, data were converted to a common format (similar to the csv CNR_DIN_WMED_20042017 data
product, Belgacem et al., 2019). This recently released product contains measurements covering the
WMED from 2004 to 2017. The data of the CNR_DIN_WMED_20042017 product have undergone a
rigorous quality control process that was focused on a primary quality check of the precision of the data
and a secondary quality control targeting the accuracy of the data. Adjustments were applied to
measurements when bias was detected.
As detailed in Table 2, we combined observations from reliable sources (covering the time period 1981-
2017), that were quality controlled according to international recommendations before being published
(Maillard et al., 2007; SeaDataNet Group, 2010). Though, these historical data collections coming from
sources different from the CNR_DIN_WMED_20042017 have been subjected to a quality check before
merging them, to eliminate the effect of any aberrant observation. The check was carried out by
computing median absolute deviations in 19 pressure classes (referring to the selected vertical resolution
of section 2.1)(0-10, 10-30, 30-60, 60-80, 80-160, 160-260, 260-360, 360-460, 460-560, 560-900, 900-
1200, 1200-1400, 1400-1600, 1600-1800, 1800-2000, 2000-2200, 2200-2400, 2400-2600, >2600 dbar).
Any value that is more than three median absolute deviation from the median value is considered a
suspected measurement.
In total, 2.35% of nitrate observations, 2.44% of phosphate observations and 2.14% of silicate
observations were removed.
**3 Methods**
3.1 Variational analysis mapping tool
Here, the **D**ata-**I**nterpolating **V**ariational **A**nalysis- **n d**imension (DIVAnd) method (Beckers et al., 2014;
Troupin et al., 2010, 2012) was used to generate the gridded fields. DIVA has been widely applied to
oceanographic climatologies, such as the SeaDataNet climatological products (Simoncelli et al., 2014,
2016; Iona et al., 2018), EMODnet chemistry regional climatologies (Míguez et al., 2019), the Adriatic
Sea climatologies by Lipizer et al. (2014) or the black Sea (Capet et al., 2014) and it was also applied to
generate the global interior climatology GLODAPv2. 2016b (Lauvset et al., 2016). It is an efficient
mapping tool used to build a continuous spatial field from discrete, scattered, irregular in situ data points
with an error estimate at each level.
The BGC-WMED gridded fields have been computed with the more advanced N-dimensional version
of DIVA, DIVAnd v2.5.1 (Barth et al., 2014) (https://doi.org/10.5281/zenodo.3627113) using Julia as





a programming language (https://julialang.org/) under the Jupyter environment (https://jupyter.org/).
The code is freely available at https://github.com/gher-ulg/DIVAnd.jl (last access: January, 2020).
DIVA is based on the variational inverse method (VIM) (Brasseur et al., 1996). It takes into account the
errors associated with the measurements and takes account of the topography/bathymetry of the study
area. The method is designed to estimate an approximated field $\varphi$ close to the observations and find the
field that minimizes the cost function $J[\varphi]$.
The cost function is defined as the misfit between the original data $d_i$, an array of $N_d$ observations, the
analysis (observation constraint term) and a smoothness term. (Troupin et al., 2010):
$$J[\varphi] = \sum_{i=1}^{Nd} \mu_i Lc^2 \big(d_i - \varphi(x_i, y_i)\big)^2 \qquad (1) \; Observation \; constraint \; term$$
$$+ \int_D \big(\alpha_2 \nabla\nabla_\varphi : \nabla\nabla_\varphi + \alpha_1 Lc^2 \nabla_\varphi . \nabla_\varphi + \alpha_0 Lc^4 \varphi^2\big) dD \qquad (2) \; Smoothness \; term$$

211                                             Eq. (1)

where Lc is the correlation length, $\nabla$ is the gradient operator, $\nabla\nabla_\varphi : \nabla\nabla_\varphi$ is the squared Laplacian of $\varphi$,
the first term (observation constraint) considers the distance between the observations and the analysis
reconstructed field, so that $\mu_i$ penalizes the analysis misfits relative to the observations. The second term
(smoothness term) measures the regularity of the  domain of interest D. This expression within the
integral remain invariant (Brasseur and Haus, 1991). $\alpha_0$ minimize the anomalies of the field itself, $\alpha_1$
minimize the spatial gradients, $\alpha_2$ penalizes the field variability (regularization). The reconstructed
fields are determined at the elements of a grid on each isobath using the cost function Eq. (1).
The grid is dependent on the correlation length and the topographic contours of the specified grid in the
considered region, so there is no need to divide the region before interpolating.
The method computes two-, three- to four-multi-dimensional analyses (longitude, latitude, depth, time).
For climatological studies, the four-dimensional extension was used on successive horizontal layers at
different depths for the whole time period.
Along with the gridded fields, DIVA yields error fields dependent on the data coverage and the noise in
the measurements (Brankart and Brasseur, 1998; Rixen et al., 2000). Full details about the approach is
provided extensively by Barth et al. (2014) and Troupin et al. (2018) in the Diva User Guide.





3.2 Interpolation parameters
DIVAnd is conditioned by topography, by the spatial correlation length (Lc) and by the signal-to-noise
ratio (SNR, λ) of the measurements, which are essential parameters to obtain meaningful results. They
are considered more in detail in the following sections.
3.2.1 Land-sea mask
A 3D dimension land-sea mask is created using the coastline and bathymetry of the General Bathymetric
Chart of the Oceans (GEBCO) 30-sec topography (Weatherall et al., 2015). The WMED is a relatively
small area which necessitates a high-resolution bathymetry to generate a mask at different depth layers.
The vertical resolution is set to 19 standard depth levels from the surface to 1500 m: 0, 5, 10, 20, 30, 50,
75, 100, 150, 200, 250, 300, 400, 500, 600, 700, 800, 1000, 1500 m, corresponding to the most
commonly used predefined levels for the sampling of seawater for nutrient analyses. The resulting fields
at each depth level are the interpolation on the specified grid. These depth surfaces are the domain on
which the interpolation is performed.
3.2.2 The spatial correlation length scale (Lc)
Lc indicates the distance over which an observation affects its neighbors. The correlation length can be
set by the user or computed using the data distribution.
For the BGC-WMED biogeochemical climatology, this parameter was optimized for the whole-time
span, and at each depth layer. The correlation length has been evaluated by fitting the empirical kernel
function to the correlation between data isotropy and homogeneity in correlations. The quality of the fit
is dependent on the number of observations (Troupin et al., 2018). The analytical covariance model used
in the fit is derived for an infinite domain (Barth et al, 2014). To assess the quality of the fit, the data
covariance and the fitted covariance are plotted against the distance between data points (Fig. 4). At 10
m, the correlation length was obtained with a high number of data points, indicating that the empirical
covariance used to estimate the covariance and the fitted covariance are in good agreement.
At some depth layers there are irregularities due to an insufficient amount of data points, making it
necessary to apply a smoothing filter/fit to minimize the effect of these irregularities. It has been tested
whether a randomly selected field analysis (nitrate data from 2006 and 2015) obtained with the fitted-
vertical correlation profile is better than the analysis with zero-vertical correlation. A skill score relative
to analysis non-fitted-vertical correlation has been computed following Murphy (1988) and Barth et
al.(2014):
$skill\ score = 1 - \dfrac{RMS^2_{no\ fit}}{RMS^2}$                                        Eq. (2)
A large difference in the global RMS between the analysis with the fitted-vertical correlation and the



analysis with non-fitted-vertical correlation used for validation was found. The test shows whether the
use of the fit in the correlation profile is improving the overall analysis or not. We found that the RMS
error was reduced from 0.696 $\mu$mol kg$^{-1}$ (analysis without fit) to 0.571 $\mu$mol kg$^{-1}$ (analysis with fit),
which means using the fitted vertical correlation profile in the analysis improves the skill by 32 %, and
the fit is improving the analysis fields.

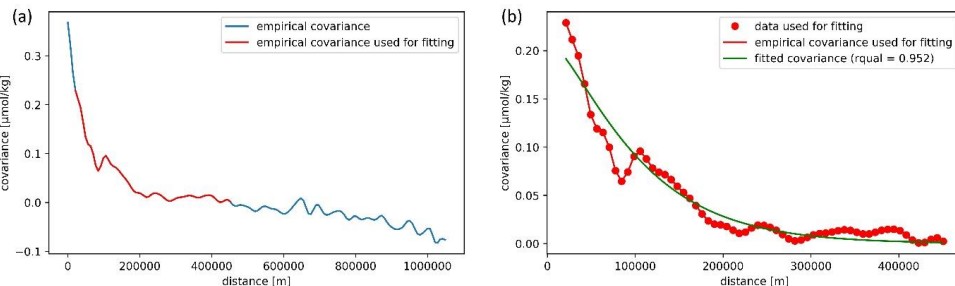


**Figure 4**. Example of the Nitrate covariance. (a) The empirical data covariance function is given in red,
curve comes from the analysis of observations within depth = 10 m, while (b) the fitted covariance curve
(theoretical kernel) is given in green.
Based on the data, DIVA performs a least-square fit of the data covariance function with a theoretical
function. Then, a vertical filter is applied and an average profile over the whole period is used (Fig. 5).
This procedure is analogous to what has been used for the EMODnet climatology and the North Atlantic
climatology, except that in EMODnet climatology, seasonally averaged profiles were used (Buga et al.,
2019) and  a monthly averaged profiles were used in North Atlantic climatology (Troupin et al., 2010).
The filter is applied to discard aberration caused by outliers or scarce observations in some layers, as
described above.
Because of the horizontal and vertical inhomogeneity of the data coverage, the analysis was based on a
correlation length that varies both horizontally (Fig. 5a) and vertically (Fig. 5b).
As expected, Lc increases with depth (Fig. 5), extending the influence area of the observation, a
consequence of the fact that variability at depth is lower and that observations in the deep layer are
scarcer (which on the other hand makes the Lc estimate more uncertain).
From the surface to 150-200 m, Lc is rather constant, while from 200 to 600 m, the horizontal Lc
increases for all nutrients.  The vertical Lc behaves similarly, for nitrate and phosphate, due to the
homogeneity of the intermediate water mass, as explained also by Troupin et al. (2010). For silicate, the
vertical Lc decreases in the intermediate depth, reaching a minimum at 500 m depth. The different
behavior of silicate could be explained by the progressive increase in concentrations from the surface to
the deep layer, compared to nitrate and phosphate vertical distribution (strong gradient between surface
depleted layer and intermediate layer). Silicate is less utilized by primary producers, and the dissolution



of the biogenic silica is slower than that of the other nutrients (DeMaster, 2002) which explain its
progressive increase towards deeper layers (Krom et al., 2014).
Below 600 m, the horizontal Lc for silicate decreases down to 1000 m, and then increases again at 1500
m. For nitrate and phosphate, a similar, but less marked, behavior is observed. The vertical Lc for all
nutrients increases progressively from 400 m to 1500 m.
Troupin et al. (2010) and Iona et al. (2018) attributed similar changes observed in Lc for temperature
and salinity to the variability of the water masses in each layer. This might also explain the changes
found in Lc for nutrients. Indeed, the concentration of nutrients in the WMED increases with depth and
is very low at the surface, which explains the constant low values of Lc in this layer.

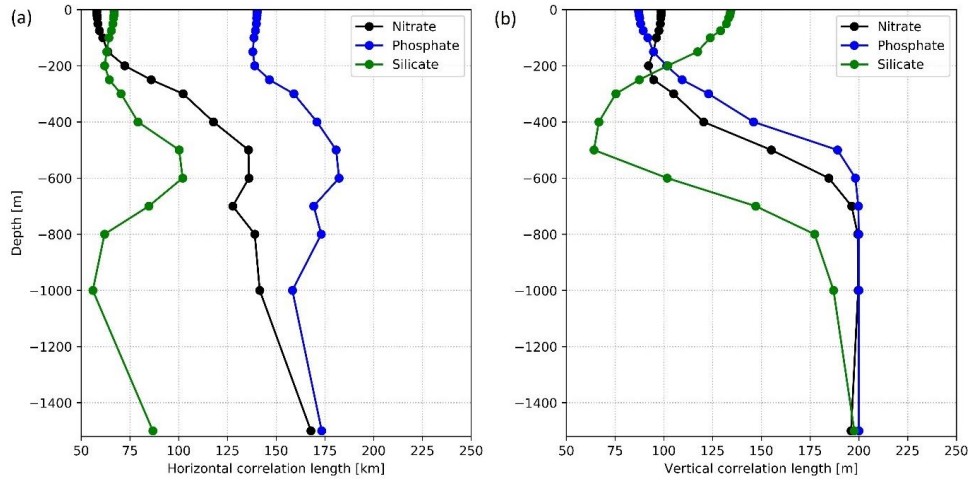


**Figure 5.** (a) Horizontal and (b) vertical optimized correlation lengths, for each nutrient (1981-2017),
as a function of depth.
3.2.3 Signal-to-Noise Ratio
The signal-to-noise ratio (SNR) is related to the confidence in the measurements. It is the ratio between
the variance of the signal and the variance of the measurement noise/error. The SNR defines the
representativeness of the measurements relative to the climatological fields, in other words it is the
confidence in the data.
It not only depends on the instrumental error but also on the fact that observations are instantaneous
measurements, and since a climatology is a long-term mean, such observations do not represent exactly
the same.
Generally, small SNR values, favor large deviations from the real measurements to give a smoother
climatological field. On the other hand, with a high SNR, DIVAnd keeps the existing observations and



interpolates between data points. The need is to find an approximation that does not deviate much from
the real observations (further details in Lauvset et al., 2016, and Troupin et al., 2010).
Following the same approach that many climatologies that used the DIVAnd method adopted, i.e.
EMODnet climatologies (available on the EMODnet chemistry portal), the Atlantic regional
climatologies (Troupin et al., 2010), the Adriatic Sea climatology (Lipizer et al., 2014) and the
SeadataNet regional climatology (Simoncelli et al., 2015), the SNR is set to a constant value (Table 1).
The analysis is performed with a predefined uniform default error variance of 0.5 for all parameters at
all depths. Three iterations are done inside DIVAnd to estimate the optimal scale factor of error variance
of the observation (following Desroziers et al., 2005). More details can be found in https://gher-
ulg.github.io/DIVAnd.jl/latest/#DIVAnd.diva3d.
Values of  SNR provided by means of a generalized cross-validation (GCV) technique (Brankart and
Brasseur, 1998) gave a large estimate of the SNR (of  the order of 22) showing a discontinuous analysis
field and patterns around the cruise transects and do not represent properly the climatological fields.
High SNR means less confidence in the observation, while we presume that the data sources used to
generate BGC-WMED climatology are consistent products.
3.3 Detection of suspicious data
Assessment of the analysis is performed by detecting outliers and suspicious data , in order to remove
observations that generate irregular interpolated fields and suspect observations that were not detected
in the data quality check of section 2.3.
The automatic check measures how consistent the gridded field is with respect to the nearby
observations by estimating the difference between a measurement and its analysis scaled by the expected
error and, based on that, a score is assigned to each observations. Data points with high scores were
considered as suspect and were removed from the analysis. Overall, 0.031%, 0.014%, 0.004% data
points, for nitrate, phosphate, and silicate, respectively, were considered inconsistent. The quality check
values that were used are available in the netCDF files of the product.
3.4 Quality check of the analysis fields
The quality of the climatology was checked against observations by estimating the mean residual and
RMS of the difference between the climatology and the observations. Averages over the entire basin
were calculated between depth levels (see section 2.3).
Residuals are the difference between the observations and the analysis  (interpolated linearly to the
location of the observations). The residuals are NaN when the observations fall outside the selected
domain for the climatology. as defined by the mask and the coordinates of the observations



The result of Fig. 6a shows nitrate residuals. From the 0 to 30 m depth, the observations and the analysis
have a high level of agreement. Between 30 and 200 m, boxplots are suggestive of larger differences.
From surface to the deep layer, the mean residual varied between -0.075 and 0.0765 $\mu$mol kg$^{-1}$. The
RMS for nitrate varied between 0.47 and 1.1 $\mu$mol kg$^{-1}$.
As for phosphate residuals (Fig. 6b), low level of agreement was found between 75 and 200 m and a
lower difference in the surface and below 250 m. The average residual varied between -0.0027 and
0.0026 $\mu$mol kg$^{-1}$. The RMS for phosphate varied between 0.037 and 0.063 $\mu$mol kg$^{-1}$.
Silicate residuals (Fig. 6c), on the other hand, seemed more homogeneous at all depth levels. The highest
level of agreement was found below 20 m and at 600 m. Overall residuals varied between -0.057 and
0.063 $\mu$mol kg$^{-1}$, while the RMS ranged between 0.567 and 0.963 $\mu$mol kg$^{-1}$.
Over the entire water column, the mean residual was around zero (0.004 $\mu$mol kg$^{-1}$ for nitrate, 0.0002
$\mu$mol kg$^{-1}$ for phosphate and 0.003 $\mu$mol kg$^{-1}$ for silicate) (Fig. 6), meaning that in general, the bias
between the observations and the analysis is small.

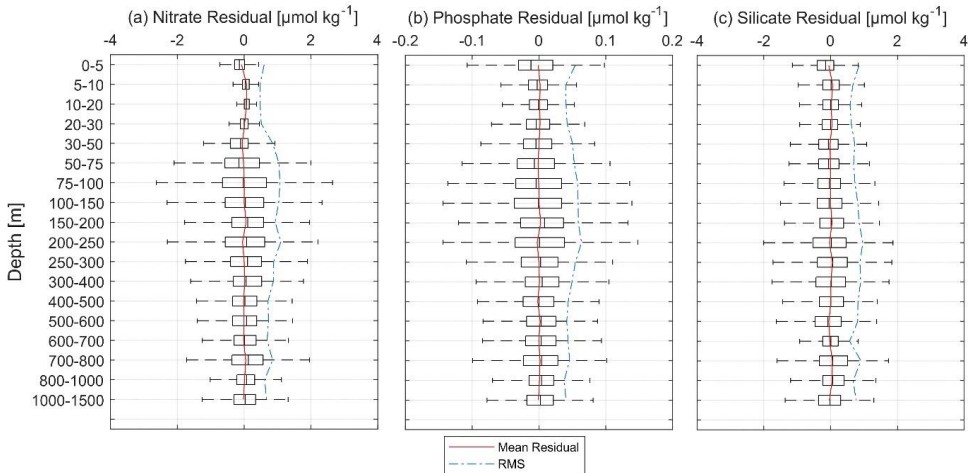


**Figure 6**. Vertical mean residuals (in red), i.e. the differences between the observations and the analysis
and the mean RMS (dashed blue) of (a) nitrate, (b) phosphate, (c) silicate.
**4 Results**
The final result consists of gridded fields of mapped climatological means of inorganic nutrients for the
periods 1981-2004, 2005-2017, and the whole period 1981-2017, produced with VIM described in
section 3, using data of section 2. Together with the gridded fields, error maps have been generated to
check the degree of reliability of the analysis.



The resulting climatologies (Table 3) are aggregated in a 4D netCDF for each nutrient and each time
period that contains the interpolated field of the variable and related information: associated relative
error, variable fields masked using two relative error thresholds (L1 and L2). The mapped climatology
is available from PANGAEA (https://doi.pangaea.de/10.1594/PANGAEA.930447, Belgacem et al.,
2021) as one folder named BGC-WMED climatology. This folder contains nine netCDF files for each
parameter and time period.
Here is an example of the analysis output found in the netCDF. Figure 7 shows the unmasked
climatological field of the mean spatial variation of nitrate, relative error field distribution, the masked
climatological field using relative error with two threshold values (0.3 and 0.5) to assess the quality of
the resulting fields.
**Table 3**. Available analyzed fields and available information in the netCDF files.

| Variable name | Field name | Description |
|---|---|---|
| Lon | Longitude | Longitude in degrees east, extent: -7 – 17.25 °E |
| Lat | Latitude | Latitude in degrees north, extent: $33.5 – 45.85$°N |
| depth | Depth | Depth in meters, 19 levels, range: 0 – 1500 m |
| nitrate/phosphate/silicate | DIVAnd analyzed climatology | Mapped climatological fields |
| nitrate_L1/phosphate_L1/ silicate_L1 | Nitrate/Phosphate/Silicate masked field level 1 | Mapped climatological fields masked using relative error threshold 0.3. |
| nitrate_L2/ phosphate_L2/ silicate_L2 | Nitrate/Phosphate/Silicate masked field level 2 | Mapped climatological fields masked using relative error threshold 0.5. |
| nitrate_relerr/phosphate_re lerr/silicate _relerr | Nitrate/Phosphate/Silicate masked relative error | Mapped relative error filed associated to the climatological field |




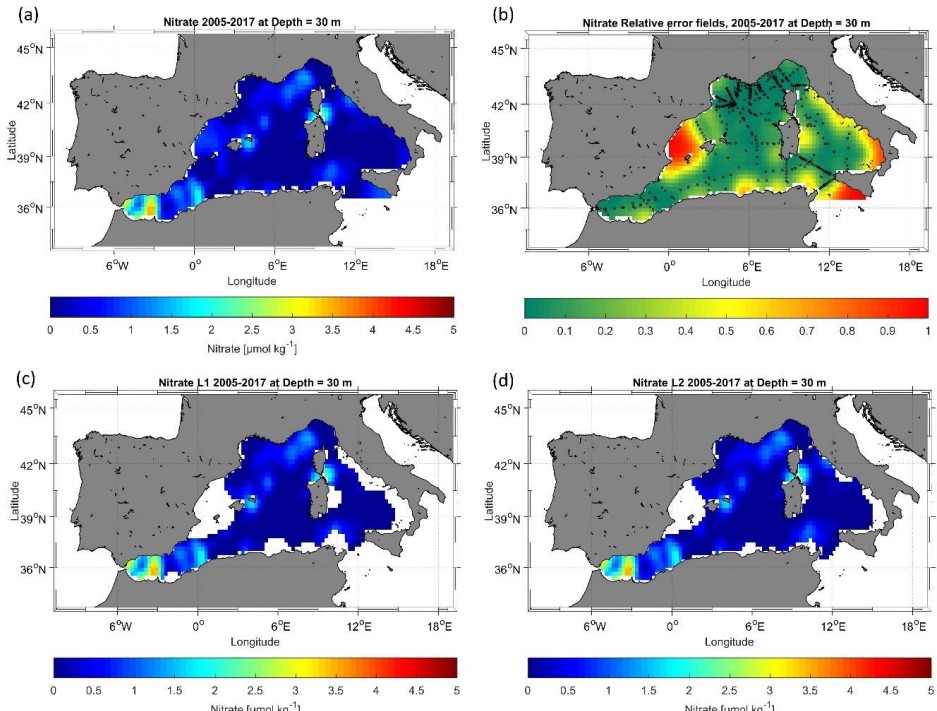


**Figure 7.** Example of nitrate analysis for the period 2005-2017 (a) unmasked analysis field, (b) relative

error field distribution with the observation in black circles, (c) masked analysis fields masked using

relative error threshold = 0.3, and (d) masked analysis fields masked using relative error threshold = 0.5.

## 4.1 Nutrient climatological distribution

A description of the spatial patterns of the dissolved inorganic nutrients across the domain and over the

entire period (1981-2017) is given. The gridded fields for nitrate, phosphate, and silicate are discussed

at three depth levels, representative of the surface (at 100 m), intermediate (at 300 m), and deep layer

(at 1500 m). The horizontal maps at the selected depths are shown in Fig. 8, while the average vertical

profiles of nutrients over the whole area are shown in Fig. 9.

### 4.1.1 Surface layer

The nitrate, phosphate and silicate mean climatological fields over 1981-2017 are presented in Fig. 8

(a, b, c) respectively. The mean surface nitrate at 100 m is about $3.58 \pm 1.16$ $\mu$mol kg$^{-1}$. Highest surface

values of nitrate concentrations are found in regions where strong upwelling or vertical mixing occurs,

such as the Liguro-Provençal basin and the Alboran Sea (see Fig. 8a).





The convection region (Gulf of Lion and Ligurian Sea) is characterized by an eutrophic regime and a
spring bloom (Lavigne et al., 2015), unlike the rest of the basin that shows low nitrate concentrations in
the surface layer ($< 4\ \mu$mol kg$^{-1}$).
Nutrient patterns in the Alboran Sea have been associated with the distinct vertical mixing that supplies
the surface layer with nutrients (Lazzari et al., 2012; Reale et al., 2020).
Indeed, the northern Alboran Sea is known as an upwelling area, where permanent strong winds enhance
the regional biological productivity (Reul et al., 2005). Nitrate distribution at 100 m presents a clear
distinction between the enriched surface regions in the WMED, under the influence of deep convection
processes, and the easternmost depleted region.
The distribution of phosphate concentration has striking similarities with that of nitrate (Fig. 8b). The
mean surface phosphate concentrations at 100 m, is $0.16 \pm 0.06\ \mu$mol kg$^{-1}$. As for nitrate, the highest
surface values are found in the Alboran Sea, Balearic Sea, Gulf of Lion and Liguro-Provençal Basin
(0.2-0.3 $\mu$mol kg$^{-1}$), while the Tyrrhenian Sea and the Algerian Sea revealed phosphate concentration
that were <0.2 $\mu$mol kg$^{-1}$. Similar patterns were observed by Lazzari et al. (2016), who argued that the
variations in phosphate are regulated by atmospheric and terrestrial inputs. It should be noted that the
maximum in the surface is found near river discharges of freshwater, like Ebro and Rhône, i.e. the largest
rivers of the WMED (Ludwig et al., 2009).
Concerning the distribution of silicate concentration, the surface layer at 100 m (Fig. 8c) followed the
same pattern as nitrate and phosphate. Over this layer the mean silicate was about $2.7 \pm 0.7\ \mu$mol kg$^{-1}$.
As for nitrate and phosphate, the highest values (3-4 $\mu$mol kg$^{-1}$), were recorded in the Alboran Sea,
Balearic Sea, Gulf of Lion and Liguro-Provençal Basin and in the southern entrance of Tyrrhenian Sea.
This surface distribution is in good agreement with the findings of Crombet et al. (2011), relating this
local silicate surface maximum to the continental input, river discharge and atmospheric deposition
(Frings et al., 2016; Sospedra et al., 2018). The spatial minima were reported in the Tyrrhenian Sea and
Algerian Sea (<3 $\mu$mol kg$^{-1}$).
## 4.1.2 Deep and Intermediate layer
At the basin scale, nitrate concentrations increase with depth (Fig. 9a), with the highest concentration
found at intermediate levels (250-500 m), ranging between 8.8 and 9.0 $\mu$mol kg$^{-1}$. In this 300 m (Fig.
8d), nitrate concentrations average is $7.2 \pm 1.06\ \mu$mol kg$^{-1}$. High values (> 6.5 $\mu$mol kg$^{-1}$) are found in
the westernmost regions (Alboran Sea, Algerian Sea, Gulf of Lion, Balearic Sea and the Liguro-
Provençal Basin), while the easternmost regions (Tyrrhenian Sea, Sicily Channel), exhibit much lower
concentrations (between 4.5 and 6.5 $\mu$mol kg$^{-1}$).
Similar features are observed in the deep layer, at 1500 m (Fig. 8a), with nitrate concentrations
increasing all over the basin, reaching on average 7.8 - 7.9 $\mu$mol kg$^{-1}$ between 1000 and 1500 m depth
(Fig. 9a).





In both layers (300 m and 1500 m), the difference between the eastern opening of the basin (Sicily
Channel) and the western side (Alboran Sea) is noticeable: the Sicily Channel and the Tyrrhenian Sea
are under the direct influence of the water masses coming from the oligotrophic EMED, which then
gradually become enriched with nutrients along its path, as found by Schroeder et al. (2020).
Phosphate concentrations at intermediate depth (see 300 m, Fig. 8e), varied between 0.12 and 0.44 $\mu$mol
kg$^{-1}$, and the horizontal map shows the same gradual decrease towards east, with the highest
concentrations in the westernmost regions and minimum values in the eastern regions ($< 0.25$ $\mu$mol kg$^{-}$
$^{1}$) .
The average vertical profile over the entire region (Fig. 9b), reveals a maximum in phosphate
concentrations between 300 and 800 m depth, related to an increased remineralization process.
In the deep layer (see 1500 m, Fig. 8h), phosphate concentration average is $0.36 \pm 0.02$ $\mu$mol kg$^{-1}$.
Generally, the deep layer is homogeneous (Fig. 9b). The difference observed between westernmost
regions and the Tyrrhenian Sea remains, though the latter demonstrate higher phosphate concentrations
($\sim 0.3$ $\mu$mol kg$^{-1}$). This variation could be due to the difference in the water masses. The IW inflow from
the EMED brings relatively young waters that are depleted in nutrients, while in the higher
concentrations in the deep layer are signatures of the older resident DW of the Tyrrhenian. The change
in the biological uptake in the intermediate source water could explain the regional variability of
nutrients. The low productivity (D'Ortenzio and Ribera d'Alcalà, 2009) and the pronounced
oligotrophic regime of EMED water (Lazzari et al., 2016) may justify the increase in nutrients in the
IW.
Silicate concentration distribution at intermediate (300 m, Fig. 8f) and deep layers (1500 m, Fig. 8i),
were as expected, showing a notable increase, compared to the surface. Here the silicate average
concentration is $5.83 \pm 0.66$ $\mu$mol kg$^{-1}$. The maximum values were observed below 800 m, $> 8.034$ $\mu$mol
kg$^{-1}$ (Fig. 4.9c). At 1500 m, silicate distribution is homogeneous  all over the basin (on average $8.35 \pm$

448    0.39).

Generally, primary producers do not require silicate for their growth as much as they need nitrate and
phosphate which explain the disparity between nutrients patterns. Furthermore, at intermediate levels,
the water is warmer than at deep levels, enhancing the dissolution rate and the progressive increase in
silicate (DeMaster, 2002). The biogenic silicate is exported to greater depths and continues to dissolve
generating inorganic silicate as it sinks to the bottom. The recycling of silicate within the deep-sea
sediments is later on redistributed by the deep currents which explain the homogenous horizontal
distribution over the entire basin.
Comparing the three nutrients at the same depth levels, at the surface (100 m), it appears that they all
show local surface maximum, depending on local events such as strong winds, local river discharge and,
vertical mixing (Ludwig et al., 2010).
In  the easternmost areas, the surface depletion in nutrients (Van Cappellen et al., 2014) is attributed to
the variation in the thermohaline properties that has impacted primary production (Ozer et al., 2017) and



the export of organic matter to intermediate and deep layers leading to the accumulation of nutrients in
these depth ranges.
The Tyrrhenian sea is not directly connected to convection regions. Here, the EMED water inflow plays
a major role. Li and Tanhua (2020) found an increased ventilation of the intermediate and deep layers
during 2001 to 2018 in the Sicily channel and a constant AOU between 2001-2016, suggesting a constant
ventilation that explain the peculiar nutrient distribution in that area. In the western side of the WMED,
intermediate and deep layers exhibit an increase in nutrients. Schroeder et al. (2020) explained this
increase in nitrate and phosphate at the intermediate layer with the increase of the remineralization rate
at these depths along the path of IW.
The deficiency of inorganic nutrients is explained by the effect of the anti-estuarine circulation, with the
IW coming from the EMED, which is known to be poor in nutrients (Krom et al., 2014; Schroeder et
al., 2020), accumulates nutrient along its path. Thus, this relative nutrient-rich Mediterranean outflow is
lost to the Atlantic Ocean.
Overall, in surface layer, circulation, physical processes, and vertical mixing increase nutrient input
while the biological pump controls the decrease.
In the deep layer, the variability is lower (standard deviation is reduced toward the bottom for all three
nutrients, see Fig.9), the deep layer accumulates dissolved organic nutrients. In the WMED, the deep
layer constitutes a reservoir of inorganic nutrients.

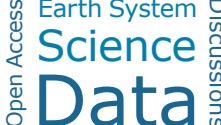

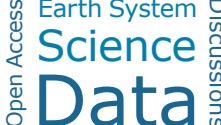

**Figure 8.** Climatological map distribution of nitrate (a. at 100 m, d. at 300 m, g. at 1500 m), phosphate (a. at 100 m, d. at 300 m, g. at 1500 m) and silicate (a. at 100 m, d. at 300 m, g. at 1500 m) for the period from 1981 to 2017.


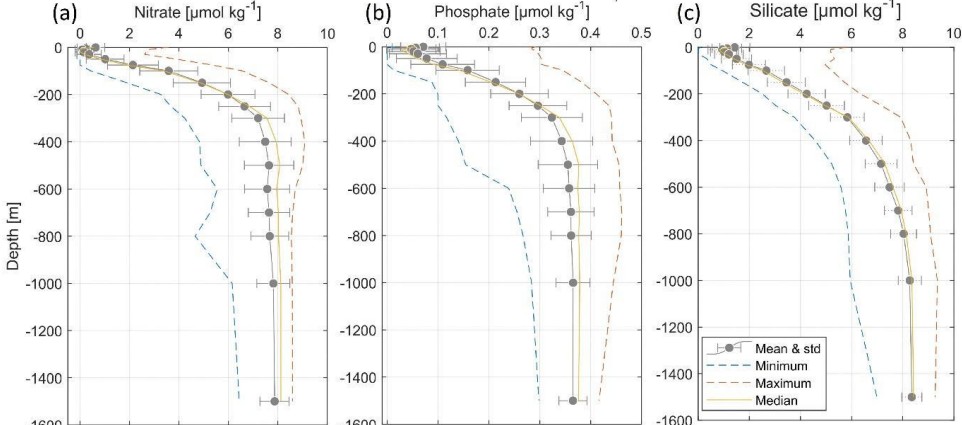


**Figure 9**. Climatological mean vertical profiles of (a) nitrate, (b) phosphate and (c) silicate concentrations in the WMED (1981-2017). Dashed blue line indicates the minimum, dashed orange line indicates the maximum, continuous yellow line indicates median profile, error bars and mean profile are in grey.

## 4.2 Error fields

The determination of the error field is important to gain insight in the confidence in the climatological results. Mostly, the error estimate depends on the spatial distribution of the observations and the measurement noise. In DIVAnd, there are different methods available to estimate the relative error associated with the analysis fields.

A climatological field is computed at several depths (19 levels in this case), for different parameters (nitrate, phosphate, and silicate in this case). Given these premises and following the approach of similar climatologies (GLODAPv2.2016b, Lauvset et al., 2016; SeaDataNet aggregated data sets products, Simoncelli et al., 2015), for the BCG-WMED the error fields were estimated using the default DIVAnd method, i.e. the "clever poor man's error approach", a less time consuming but efficient computational approach. According to Beckers et al. (2014) who also provide details about the mathematical background of the error fields computation, this method appropriately represents the true error and provides a qualitative distribution of the error estimate. This estimate is used to generate a mask over the analysis fields. Two error thresholds were applied (0.3 (L1) and 0.5 (L2)). Fig.7b., show the main error that occurs in region void from measurements. An example of the analysis masked with the error thresholds output is shown in Fig.7c (L1) and Fig.7d (L2). The associated error fields with the analysis fields are integrated in the data product.





4.3 Comparison with other biogeochemical data products
In this section a comparison of the BGC-WMED product with the most known global and/or regional
climatologies, that are frequently used as reference products for initializing numerical models, is done.
Specifically, the analyzed fields are compared to the reference data products WOA18 (Garcia et al.,
2019) and the reanalysis of the Mediterranean Sea biogeochemistry, medBFM, a CMEMS product
(Teruzzi et al., 2019). Since the products used for inter-comparison were not originated from the same
interpolation method, not for the same time period and with different spatial resolution, here the
comparison is mostly targeted on the general patterns of nutrients in the region.
Comparisons are carried out between horizontal maps (Fig.10-11-12), as well as along a vertical
longitudinal transect (Fig.14-15). In addition, following Reale et al. (2020), the first 150 m have been
evaluated (Fig.13), since this is a depth level with a representative amount of in situ observations in all
three products. The evaluation is based on the estimation of horizontal average, on BGC-WMED
climatology, the medBFM biogeochemical reanalysis and the WOA18 climatology by subregion. i.e. a
spatial subdivision made according to Manca et al. (2004).
Products have different grid resolution, thus to compare between them, the BGC-WMED new
climatological data product (at 0.25° × 0.25°) for periods 1981-2017 and 2005- 2017 and the medBFM
biogeochemical reanalysis (at 0.063° × 0.063°) (Teruzzi et al. 2019)
(https://doi.org/10.25423/MEDSEA_REANALYSIS_BIO_006_008) for the period 2005- 2017, are
regridded on the WOA18 (1° × 1°) grid using nearest neighbor interpolation. The regridding is computed
at all depth levels of the different products. The MedBFM reanalysis climatological mean was computed
for the period 2005-2017 prior the interpolation.
4.3.1 Comparison with WOA18 at 150 m
Fig. 10-11-12 show the analysis at the 150 m depth surface for the three nutrients. The BGC-WMED
(1981-2017) product reveals detailed aspects of the general features of nitrate (Fig. 10.a), phosphate
(Fig. 11a) and silicate (Fig.12a).
For the three nutrients, the new product reproduces patterns similar to the WOA18 all over the region.
It shows well-defined fields and higher values of nitrate and phosphate concentrations. In the new
product, nitrate concentrations varied between 2.31 -7.3 $\mu$mol kg$^{-1}$ the WOA18 values were 2.19 - 5.99
$\mu$mol kg$^{-1}$.Phosphate ranges were similar between the two products between (0.092- 0.35 $\mu$mol kg$^{-1}$
(BGC-WMED) and 0.095 - 0.35 $\mu$mol kg$^{-1}$ (WOA18)). Likewise, Silicate range values at 150 m were
not different (2.07 - 4.99 (BGC-WMED) and 1.57 - 5.75 $\mu$mol kg$^{-1}$(WOA18)).
The average RMS difference (RMSD) calculated from the difference between the WOA18 and BGC-
WMED all over the region at 150 m is about 1.14 $\mu$mol kg$^{-1}$ nitrate (Fig. 10c), 0.055 $\mu$mol kg$^{-1}$ for



phosphate (Fig. 11c) and 0.91 $\mu$mol kg$^{-1}$ for silicate (Fig. 12c). Overall, the RMS error values  were low
indicating limited a disparity between the two products.
The difference field for every grid point reflects this discrepancy and shows areas with limited
agreement between the two products, that can have a difference >2 $\mu$mol kg$^{-1}$ for nitrate (Fig. 10c), >0.1
$\mu$mol kg$^{-1}$for phosphate (Fig. 11c), >1.5 $\mu$mol kg$^{-1}$for silicate (Fig. 12c). This dissimilarity is also noted
with the low r$^2$ (Fig. 13) (0.34, 0.20, 0.095 for nitrate, phosphate, and silicate respectively)
The distribution of the surface nitrate concentrations (at 150m) (Fig. 10a) of the new product is similar
to that shown in WOA18 (Fig. 10b). The largest difference between the two products occurs in northwest
areas and in the Alboran Sea (Fig. 10c), areas of higher concentrations, a more nutrient rich surface
water as described in section 4.1. The difference is pronounced in these regions likely because the new
product holds more in situ observations than the WOA18 in the WMED.
Phosphate surface concentrations (Fig. 11) show similar differences as nitrate. The largest difference
with the surface phosphate of the WOA18 is found in the Alboran Sea, Northern WMED and Sicily
region (Fig. 11c).
As for silicate, the surface distribution shows large differences (Fig. 12c). The highest values are
observed in the northwest area of the new product, and in the Alboran Sea in the WOA18 climatology ,
this again accounts for the data coverage difference.

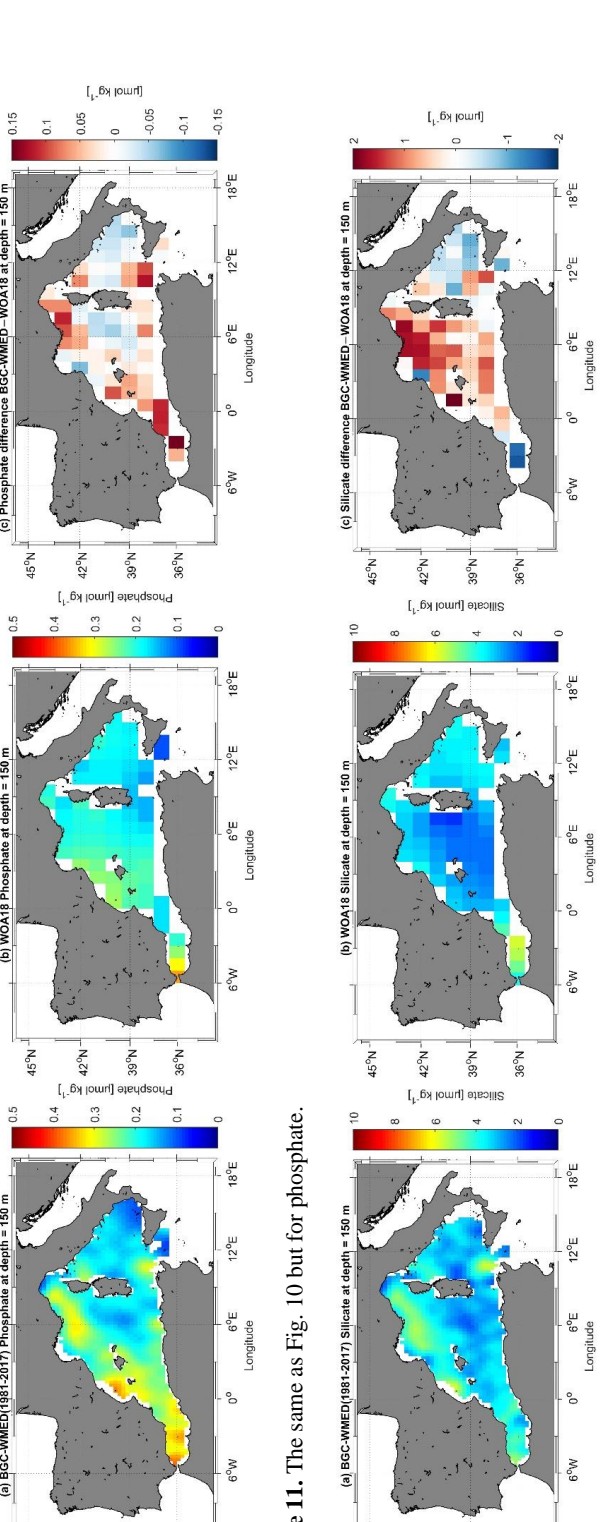

**Figure 10.** (a) BGC-WMED (1981-2017) nitrate climatological field at 150 m depth (0.25° resolution); (b) WOA18 nitrate climatological field at 150 m depth (1° resolution); (c) difference between BGC-WMED and WOA18 nitrate fields at 150 m (1° resolution).

**Figure 11.** The same as Fig. 10 but for phosphate.

**Figure 12.** The same as Fig. 10 but for silicate.

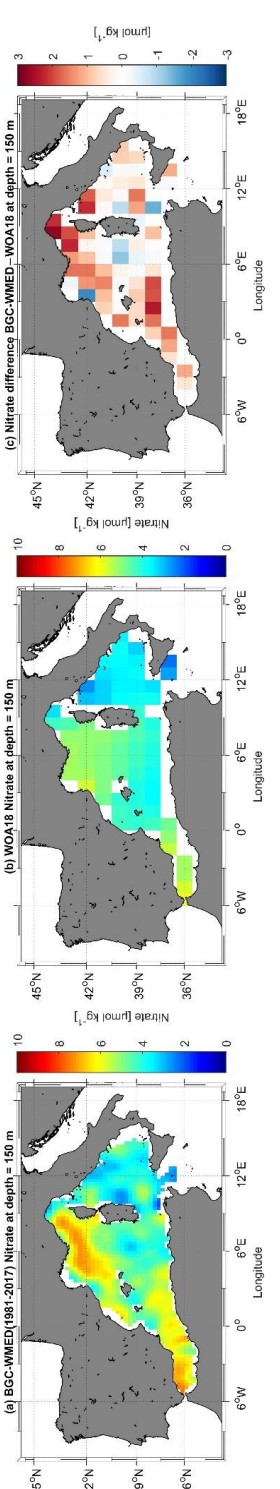

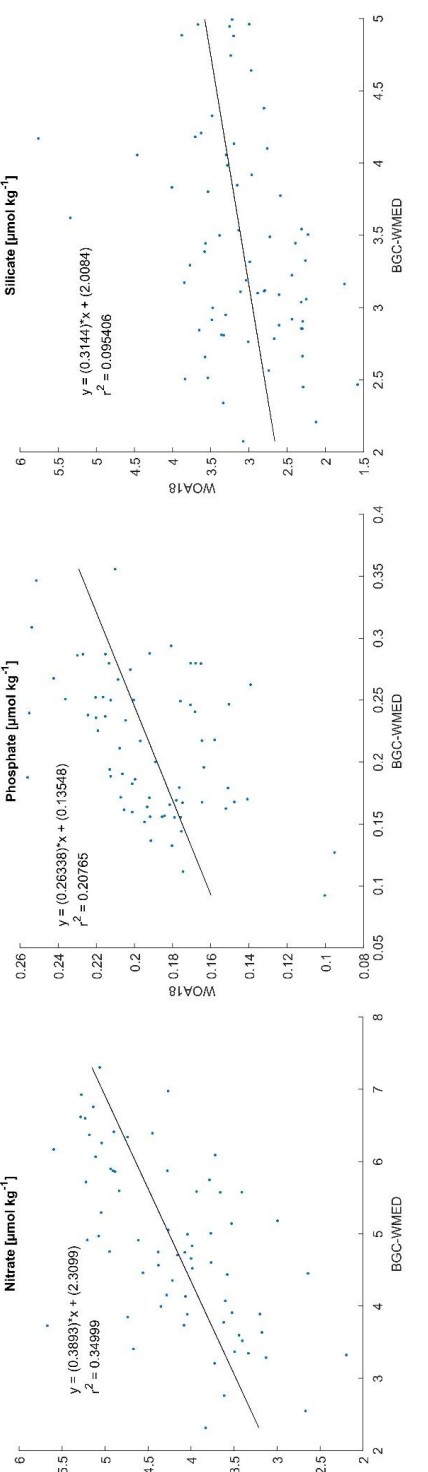

**Figure 13**. Scatterplot showing the WOA18 data as a function of the BCG-WMED climatology at 150 m with the regression line.





4.3.2 Regional horizontal comparison above 150 m average nutrient concentrations

The inorganic nutrient mean concentrations resulting from the climatology of this work (period 2005-2017), and from both the medBFM reanalysis product and the WOA18 are compared in the upper layer of 12 subregions of the WMED (in Table 4 and Fig. 14).

Results show a general agreement between BGC-WMED and the other two products in some subregions, nonetheless, there are some differences as shown in section 4.3.1.

Upper layer nitrate average concentrations (Fig. 14a) are decreasing eastward, from the Alboran Sea (DS1) to the Algerian basin (DS3, DS4) and the Balearic Sea (DS2). The western part of the basin is an area under the direct influence of the inflowing Atlantic surface waters, where nitrate is known to be present in excess compared to phosphate probably due to atmospheric $N_2$ input (Lucea et al., 2003). In the DS1, BGC-WMED nitrate levels are lower than the WOA18 nitrate levels while in DS3, DS2 and DS4 the average nitrate concentrations are similar to the WOA18.

From the Algerian basin (DS4, DF1) to Liguro-Provençal (DF3) regions, there is an increase in the average nitrate in all products, this is the south-north gradient. Some difference arises, where the new product is lower than the WOA18.

In the eastern regions, the lowest average concentrations of the WMED are found. Here, the difference between products is smaller, with medBFM reanalysis being lower than the new product and the WOA18.

As for phosphate (Fig. 14b), known to be the limiting nutrient of the WMED, because it is rapidly consumed by phytoplankton (Lucea et al., 2003), its average levels are low in DS1, DS3, DS2 and DS4, in WOA18, medBFM reanalysis and BGC-WMED. The latter did not agree well with the other products in DS2, where it was slightly higher. Phosphate average concentrations slightly increase in DF1, DF2 and DF3 in all three products. The increase is explained by the vertical mixing process occurring in the northern WMED.

Upper surface phosphate concentrations average start to decrease progressively through the Ligurian East (DF4), Tyrrhenian Sea (DT1, DT3), Sardinia Channel (DI1) and Sicily Channel (DI3).The BGC-WMED was in agreement with medBFM reanalysis in those subregions aside from concentrations in DI3, where the new product showed higher levels.

The BGC-WMED climatology shows reasonable agreement in the upper average concentrations of nitrate and phosphate that are similar in order of magnitude to the other products (Fig. 14). The difference with the WOA18 resides in the wider temporal window of the observation (starting from 1955). The new climatology in some subregions has a better spatial coverage of in situ observation than the WOA18 (Garcia et al., 2019) and the medBFM reanalysis (Teruzzi et al., 2019).





On the other hand, the average silicate (Fig. 14c) of the new product and the WOA18 varied between
regions. Significant difference is found between the two products in DS2, DS4, DF1, DF2, DT1, DT3,
DI1 and DI3, while in DS1, DS3 and DF4 mean silicate is consistent between the two products.
Overall, the three products show strongly similar features between regions (similar curve shape).

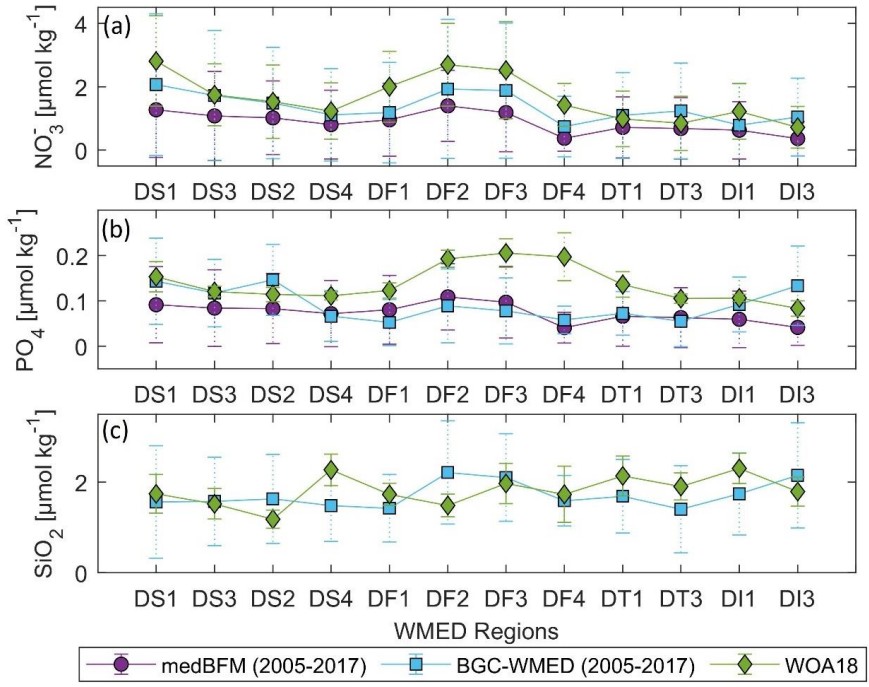


**Figure 14.** Nutrient average concentrations and standard deviation comparison in the upper 150 m
(values in Table 4).











**Table 4.** Nutrient average concentrations and standard deviation in the upper 150 m. All products were
interpolated on 1° grid resolution (see Figure S2 (Belgacem et al., 2020)).

| Subregion/ Coverage | Data product | Nitrate | Phosphate | Silicate |
|---|---|---|---|---|
| *DS1- Alboran Sea* (35°N– 37.3°N, -6°E– -1°E) | medBFM | 1.27(±1.4) | 0.09(±0.08) | - |
| | BGC-WMED | 2.06(±2.2) | 0.14(±0.09) | 1.56(±1.2) |
| | WOA18 | 2.81(±1.4) | 0.15(±0.03) | 1.74(±0.4) |
| *DS3- Algerian West* (35.36°N– 38.3°N, -1°E– 4.3°E) | medBFM | 1.07(±1.4) | 0.08(±0.08) | - |
| | BGC-WMED | 1.72(±2.05) | 0.11(±0.07) | 1.57(±0.9) |
| | WOA18 | 1.74(±0.9) | 0.12(±0.01) | 1.52(±0.3) |
| *DS2- Balearic Sea* (38.3°N– 42°N, -1°E–4.3 °E) | medBFM | 1.02(±1.1) | 0.08(±0.07) | - |
| | BGC-WMED | 1.48(±1.7) | 0.14(±0.07) | 1.63(±0.9) |
| | WOA18 | 1.53(±1.1) | 0.11(±0.01) | 1.18(±0.2) |
| *DS4- Algerian East* (36.3°N– 39.18°N, 4.3°E– 8.24°E) | medBFM | 0.80(±1.08) | 0.07(±0.07) | - |
| | BGC-WMED | 1.11(±1.4) | 0.06(±0.05) | 1.48(±0.7) |
| | WOA18 | 1.23(±0.8) | 0.11(±0.009) | 2.27(±0.3) |
| *DF1- Algero-Provençal* (39.18°N– 41°N, 4.3°E– 9.18°E) | medBFM | 0.96(±1.15) | 0.08(±0.07) | - |
| | BGC-WMED | 1.18(±1.5) | 0.05(±0.05) | 1.42(±0.7) |
| | WOA18 | 2.00(±1.1) | 0.12(±0.01) | 1.73(±0.2) |
| *DF2- Gulf of Lion* (42°N–43.36°N, 1°E–6.18°E) | medBFM | 1.39(±1.19) | 0.10(±0.07) | - |
| | BGC-WMED | 1.92(±2.1) | 0.08(±0.08) | 2.21(±1.1) |
| | WOA18 | 2.68(±1.3) | 0.19(±0.01) | 1.48(±0.2) |
| *DF3- Liguro-Provençal* (41°N– 45°N, 6.18°E– 9.18°E) | medBFM | 1.18(±1.2) | 0.09(±0.07) | - |
| | BGC-WMED | 1.88(±2.1) | 0.07(±0.07) | 2.10(±0.9) |
| | WOA18 | 2.52(±1.5) | 0.20(±0.03) | 1.97(±0.4) |
| *DF4- Ligurian East* (42.48°N–45°N, 9.18°E– 11°E) | medBFM | 0.37(±0.4) | 0.04(±0.03) | - |
| | BGC-WMED | 0.74(±0.9) | 0.05(±0.03) | 1.59(±0.5) |
| | WOA18 | 1.42(±0.6) | 0.19(±0.05) | 1.73(±0.6) |
| *DT1- Tyrrhenian North* (39.18°N–42.48°N, 9.18°E– 16.16°E) | medBFM | 0.71(±0.9) | 0.06(±0.06) | - |
| | BGC-WMED | 1.09(±1.3) | 0.07(±0.04) | 1.69(±0.8) |
| | WOA18 | 0.98(±0.8) | 0.13(±0.02) | 2.13(±0.4) |
| *DT3- Tyrrhenian South* (38°N– 39.18°N, 10°E– 16.16°E) | medBFM | 0.68(±0.96) | 0.06(±0.06) | - |
| | BGC-WMED | 1.23(±1.5) | 0.05(±0.05) | 1.40(±0.9) |
| | WOA18 | 0.84(±0.8) | 0.10(±0.01) | 1.90(±0.2) |
| *DI1- Sardinia Channel* (36°N– 39.18°N, 8.24°E– 10°E) | medBFM | 0.62(±0.9) | 0.05(±0.06) | - |
| | BGC-WMED | 0.78(±1.3) | 0.09(±0.06) | 1.74(±0.9) |
| | WOA18 | 1.22(±0.8) | 0.10(±0.007) | 2.3(±0.30) |
| *DI3- Sicily Channel* (35°N– 38°N, 10°E–15°E) | medBFM | 0.36(±0.5) | 0.04(±0.03) | - |
| | BGC-WMED | 1.04(±1.2) | 0.13(±0.08) | 2.15(±1.1) |
| | WOA18 | 0.72(±0.6) | 0.08(±0.01) | 1.79(±0.3) |

4.3.3 Regional vertical comparison of nitrate and phosphate concentrations
As the last step in the comparison between the different products, it is investigated how the new
climatology represents the vertical distribution by comparing the new climatological values for the
period 2005-2017 with the medBFM reanalysis and the WOA18.
We extracted data values along a longitudinal transect across the Algerian basin in the west-east
direction (Fig. 15). The transect was selected according to previous studies (D'Ortenzio and Ribera



d'Alcalà, 2009; Lazzari et al., 2012; Reale et al., 2020) and since the Easternmost part of the domain is
showing markedly features, a transect across the Tyrrhenian Sea is extracted as well (Fig. 15). Silicate
is not included as it was not represented in the medBFM model.
Vertical sections of nitrate and phosphate in the Algerian Sea show a common agreement between
products about the main patterns found along the water column, i.e. the nutrient depleted surface layer
and the gradual increase toward intermediate depths, we note as well the west to east decreasing gradient
in the three products, yet, there are some inequalities.
Below 100 m, there is a significant difference between products and a poor qualitative agreement.
Nitrate distribution is dominated by the nutrient enriched IW, with high values ($>7 \, \mu$mol kg$^{-1}$) increasing
from east to west (Fig. 15). Phosphate shows similar patterns in the surface layer, exhibiting very low
concentration in the surface layer and a progressive increase down to 300 m ($> 0.35 \, \mu$mol kg$^{-1}$) noted
also in the WOA18. The reanalysis showed a more smoothed field, below 100-300 m, with phosphate
concentration between 0.20 and 0.30 $\mu$mol kg$^{-1}$. The highest values for phosphate were found below 250
m from 0°E to 3°E in the new product. The BCG-WMED transect define very well the different depth
layers, the upper intermediate layer is rich with nutrient concentration with $> 8 \, \mu$mol kg$^{-1}$ for nitrate
(BGC-WMED) and $>0.35 \, \mu$mol kg$^{-1}$ for phosphate (BGC-WMED and WOA18).
The vertical section along the Tyrrhenian Sea (Fig. 15) also shows a decrease from west to east in nitrate
concentrations. The same gradient is found also in phosphate in agreement with nutrient distribution
shown from the WOA18. From the section of the medBFM reanalysis, it is not easy to identify the west-
east gradient that we mentioned before. It could be suggested that the model under-estimate the vertical
features in the Eastern (Tyrrhenian Sea: 100-300 m, nitrate vary between 1.4 and 4.2 $\mu$mol kg$^{-1}$,
phosphate between 0.13 and 0.20 $\mu$mol kg$^{-1}$) and western part (Algerian basin: 100-300 m, nitrate vary
between 2.1 and 5.4 $\mu$mol kg$^{-1}$, phosphate between 0.15 and 0.255 $\mu$mol kg$^{-1}$). These values are lower
than the ones found in the BGC-WMED (Tyrrhenian Sea: 100-300 m, nitrate range between 3 to 6 $\mu$mol
kg$^{-1}$, as for phosphate values oscillate between 0.10-0.27 $\mu$mol kg$^{-1}$;Algerian basin: 100-300 m, nitrate
range between 3.6 to 8 $\mu$mol kg$^{-1}$, as for phosphate values oscillate between 0.18-0.36 $\mu$mol kg$^{-1}$).
While the WOA18 reproduce similar patterns as the new climatology (Tyrrhenian Sea: 100-300 m,
nitrate vary between 1.8 and 5.7 $\mu$mol kg$^{-1}$, phosphate between 0.33 and 0.20 $\mu$mol kg$^{-1}$) and western
part (Algerian basin: 100-300 m, nitrate vary between 2.8 and 6.8 $\mu$mol kg$^{-1}$, phosphate between 0.16
and 0.34 $\mu$mol kg$^{-1}$).
The products illustrate the nutrient-poor water in the eastern side (Tyrrhenian Sea) and the relatively
nutrient-rich water found in the western transect (Algerian basin).



641 The BGC-WMED product capture details in Fig. 15 about the longitudinal gradient in nitrate and
642 phosphate, along the water column where nutrient sink deeper from west to east as previously seen in
643 Pujo-Pay et al. (2011) and Krom et al. (2014), an increased oligotrophy from west to east with higher
644 concentrations in the two nutrients in the western side of the section and a more oligotrophic character
645 toward east.

646 The differences between products could be explained by the difference in the data coverage, time span
647 and the difference in methods used to construct the climatological fields.

648 The variability in nitrate and phosphate fields along the transect extracted from the BGC-WMED reflects
649 the high resolution of the product allowing the screening of vertical structure controlling nutrient
650 contents. Based on a visual comparison, the new product is able to reproduce similar patterns as to the
651 WOA18 and to a lesser extend the medBFM reanalysis.

652 Fig. 16 examines the vertical difference of nitrate and phosphate concentration for the BGC-WMED
653 with the medBFM reanalysis along the Algerian basin (Fig.16a, nitrate; Fig.16b, phosphate) and
654 WOA18 (Fig.16c, nitrate; Fig.16d, phosphate).

655 The vertical section shows a strong agreement at the surface for nitrate between the BGC-WMED and
656 the medBFM reanalysis (Fig. 16a), while the vertical difference with WOA18 demonstrates that nitrate
657 values in the new product are lower than the WOA18 at 50- 75 m (Fig. 16c).

658 The difference increases with depth, below 100 m, the BGC-WMED nitrate climatology is higher than
659 the medBFM with a difference ranging between 0.6 and 2.4 $\mu$mol kg$^{-1}$, similar observation is noted in
660 the WOA18 (Fig. 16c).  In Fig.16a and Fig.16c, we identify patterns in the vertical structure of nitrate
661 in the eaten portion of the transect.

662 Regarding phosphate, differences between the new climatology and the medBFM reanalysis are noted
663 (Fig. 16b) where the BGC-WMED show high concentrations in the first 100 m and between 150 m and
664 300 m (differences of 0.02 - 0.08 $\mu$mol kg$^{-1}$), this difference decreases at 100-150 m. At the eastern
665 portion of the transect (6°E to 7.5°E), we find an agreement between the two products.

666 Conversely, the vertical sections of the differences between BGC-WMED and WOA18 in phosphate
667 (Fig.16 d) show similarities, with the new product being lower than the WOA18 in the first 50 m. Large
668 difference is found on both sides of the transect below 100 m, while in the center of the transect, the
669 difference in phosphate is reduced to 0-0.02 $\mu$mol kg$^{-1}$.

670 Fig.17 compares the vertical difference of nitrate and phosphate along the Tyrrhenian Sea transect. In
671 general, the difference transect in the Tyrrhenian Sea shows similar features  with medBFM reanalysis
672 and the WOA18 as in Algerian basin. Fig.17d captures the west to east gradient in phosphate. The
673 WOA18 overestimate phosphate in the surface layer.
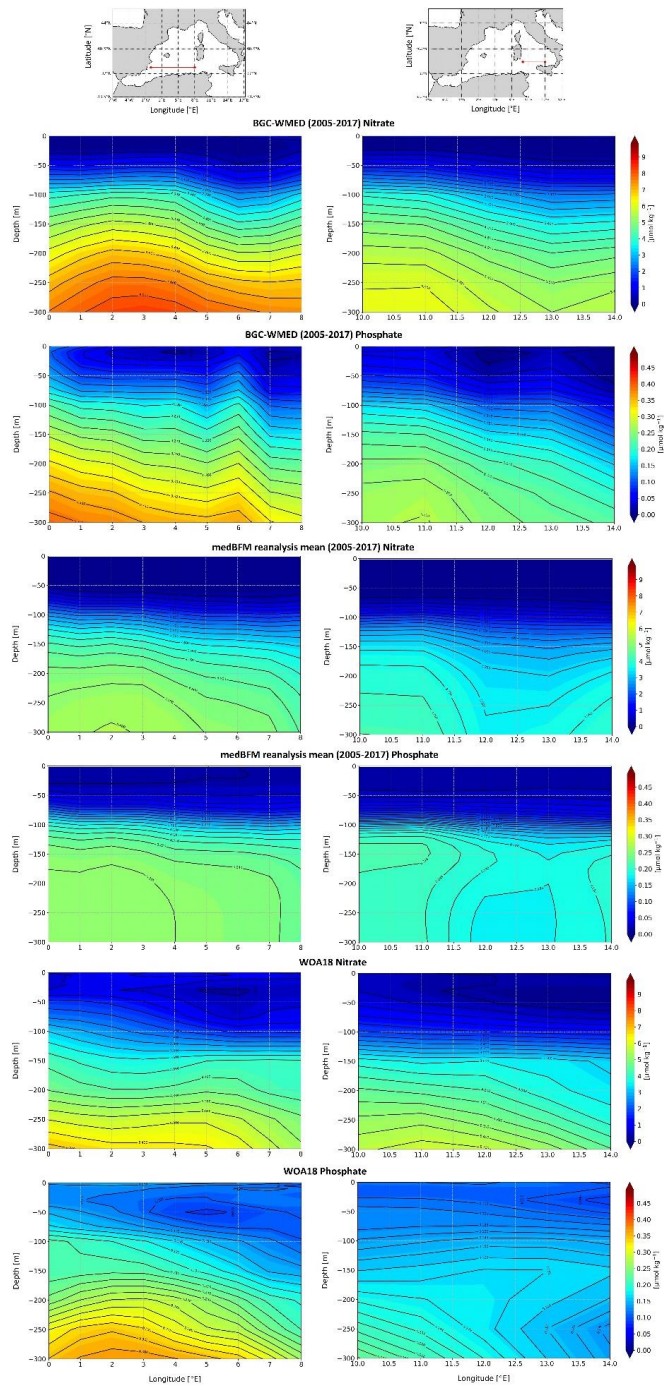






**Figure 15.** Vertical distribution of nitrate and phosphate from the Algerian basin and Tyrrhenian Sea.
Colors show the gridded values from the three different products: BGC-WMED, medBFM reanalysis
(Teruzzi et al.,2019) and the WOA18 (Garcia et al., 2019).



Based on the new climatology comparison with the WOA18 and the reanalysis, it is concluded that the
new product is consistent with the main features of previous products and show the large-scale patterns
and underline well the characteristics of the water mass layers.
The study also provides an examination of the nitrate and phosphate distributions along a longitudinal
transect across the Algerian Basin (Western WMED) and across the Tyrrhenian Sea (Eastern WMED).
We have shown that the western basin is relatively high in nutrient compared to the Eastern basin. The
increased oligotrophic gradient from west to east could be attributed to the difference in the
hydrodynamic patterns related to the water mass specific properties that are affected by the EMED and
the Atlantic ocean inflows, and to the local sources of nutrients (Ribera d'Alcalà et al., 2003; Schroeder
et al., 2010). Study of Crispi et al. (2001) inferred to the biological activity that is responsible for the
oligotrophic gradient.

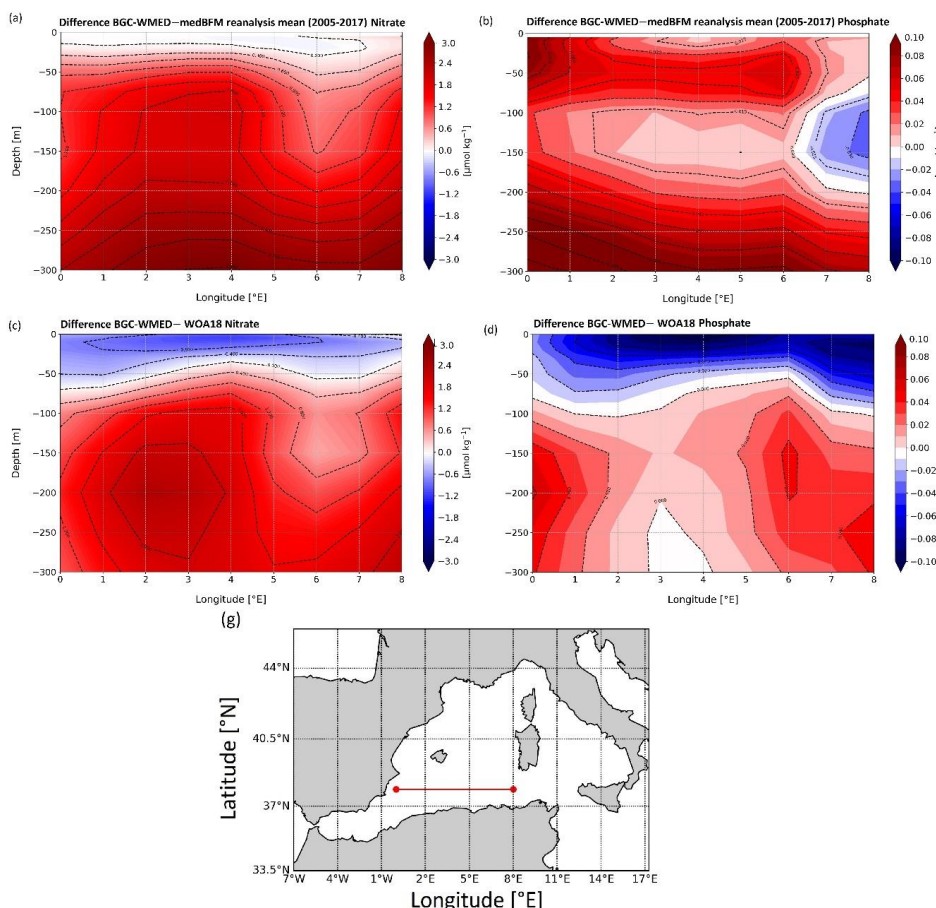


**Figure 16**. Difference of vertical section from the Algerian basin between BGC-WMED and medBFM
(a. nitrate, b. phosphate), BGC-WMED and WOA18 (c. nitrate, d. phosphate), with dashed contour lines
and labels.

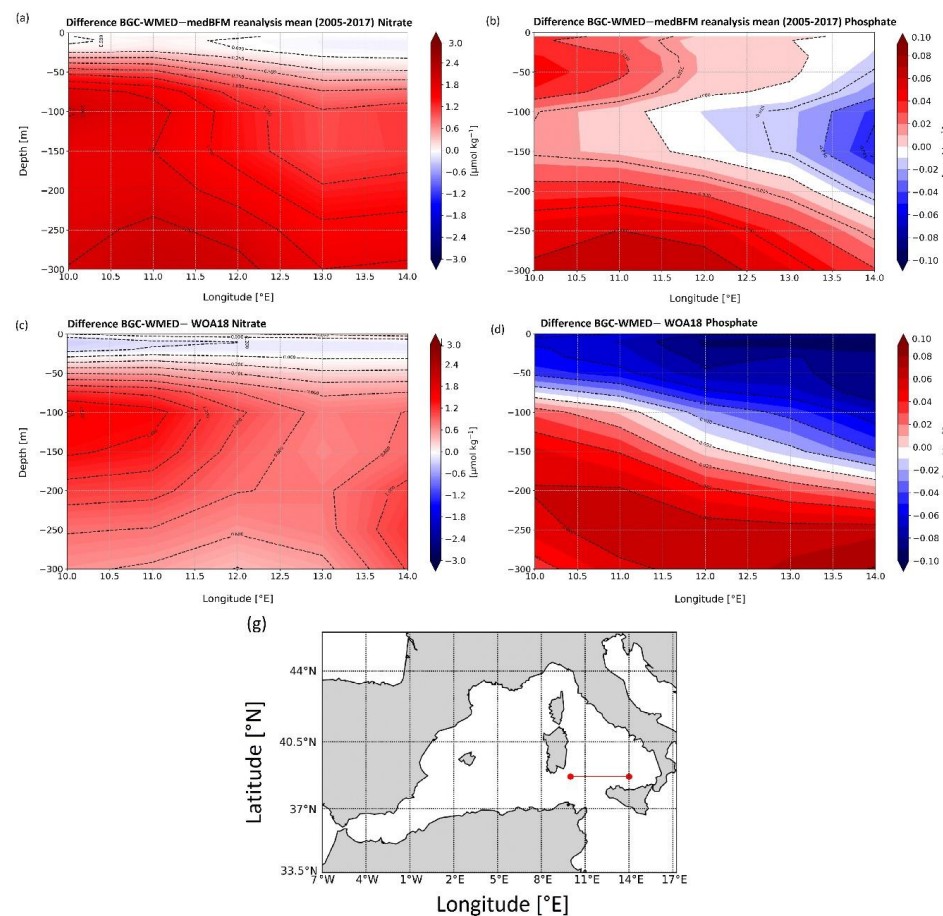

**Figure 17.** Same as Fig.16 but for the vertical section from the Tyrrhenian Sea.

## 4.4 Temporal comparison: 1981-2004 vs 2005-2017

In this section, we compare between two climatological periods (1981-2004 vs 2005-2017). The distinction between the two period was based on the occurrence of the western Mediterranean transition (WMT) that started in 2004/05, during which there was a progressive increase in temperature and salinity of the IW that led to important deep convection events, substantially increasing the rate of DW formation between 2004 and 2005 (Schroeder et al., 2016).

The result of this climatological event was that a newly generated DW, denser, saltier, and warmer than the old WMDW, filled up the WMED. The new WMDW propagated east toward the Tyrrhenian Sea and west toward the Alboran Sea and Gibraltar (Schroeder et al., 2016).

A recent study of Li and Tanhua (2020) demonstrated an enhanced ventilation in the WMED deep layers despite the continuous overall increase in temperature (Bindoff et al., 2007), salinity and density of



intermediate and deep layers after the WMT (Schroeder et al., 2016; Vargas-Yáñez, 2017). An increased
ventilation means a DW renewal (Schroeder et al., 2016; Tanhua et al., 2013) subsequently a well
oxygenated waters, implying an increase in the decomposition of the sinking organic matters into
inorganic nutrients, thus causing changes of biogeochemical cycles (Shepherd et al., 2017). What
happened in the WMED was not a permanent continuous event, since DW formation faded during the
years 2006 and 2007, to restart again in 2008 (Li and Tanhua, 2020). In this section, we investigate the
possible impact of WMT on biogeochemical characteristics at different depth levels (with a focus on
nitrate, phosphate and silicate regional distribution and patterns).
We considered depth levels that represent the usual three layers: the surface (100 m; Fig.18a-19a-20a),
intermediate (300 m; Fig.18b-19b-20b) and deep layers (1500 m; Fig.18c-19c-20c).
The WMED surface layer is dominated by the AW coming through the Alboran Sea, a permanent area
of upwelling (García-Martínez et al., 2019), where there is a continuous input of elements from the layer
below to the surface (Fig. 18a-19a-20a). Nitrate increased after WMT (Fig. 18d-19d-20d) by +0.4137
$\mu$mol kg$^{-1}$ (Fig.1Sa). The largest difference between the two periods reached >+2 $\mu$mol kg$^{-1}$ in Sardinia
Channel and the Alboran Sea that was explained by the favorable conditions for nitrogen fixation as
discussed in Rahav et al. (2013), revealing also that nitrogen fixation rate increased from east-to-west.
Phosphate and silicate on the other hand described a decrease at 100 m (Fig. A1a) with about -0.021 and
-0.1365 $\mu$mol kg$^{-1}$ in average, respectively. Large change is noticed in the southern Alboran Sea, Sardinia
channel and Balearic Sea.
The surface layer exhibits an irregular distribution since it is subjected to seasonal variability. We found
and increase in all nutrients at 300 and 1500 m with a maximum identified at intermediate depth in both
nitrate and phosphate which is explained by the remineralization of organic matter along the path of the
IW. The latter flows westward (from the Levantine to the Atlantic Ocean). Its content in nutrients
increases (relatively to the conditions in the EMED) with age (Schroeder et al., 2020). It arrives to the
Tyrrhenian Sea, where in Fig.18b-19b-20b (at 300 m depth), we identify a nutrient-depleted intermediate
layer. At this depth level, we observe a gain in the three nutrients after WMT (Fig.18e-19e-20e). In
average, the difference between the two periods (pre/post-WMT) for nitrate, phosphate, and silicate, is
around +0.8648, +0.0068 and +0.2072 $\mu$mol kg$^{-1}$ (Fig. A1b), respectively.
A similar increase after WMT in the deep layer (1500 m), is also found for nutrient concentrations (Fig.
18f, 19f, 20f) in the magnitude of +0.753 for nitrate, +0.025 for phosphate, and +0.867 for silicate (Fig.
A1c), which highlights an increase in the downward flow of organic matter remineralization that is
supplying the existing pool.

This increase is also illustrated in the climatological mean vertical profile of Fig. 21 in the three nutrients.
Nitrate displays a notable vertical difference to the pre-WMT period below 200 m (Fig.21a). Phosphate
difference between the two-time period is larger below 400 m (Fig. 21b). Silicate was different than





nitrate and phosphate. It increases progressively with depth (Fig.20c) and  demonstrated an enrichment
of the DW compared to the 1981-2004 period (Fig. 21c). The maximum values are found in the deep
layer, due to the low remineralization rate. With the warming climate, biogenic silicate tends to dissolve
faster which explain the high concentrations all over the basin even the Tyrrhenian Sea after the WMT.
According to Stöven and Tanhua (2014), the impressive volume of the newly formed DW during 2004
and 2006, ventilated the old DW decreasing its age, meaning that the WMT could have led to the
lowering of the WMED deep layer pool in nutrient as it was pointed out by Schroeder et al. (2010).
However, we did not observe this decrease in the climatological analysis after the WMT. It might be
due to the temporal variability of the deep convection intensity, since a decrease has been recorded in
the Gulf of Lion between 2007 and 2013 (Houpert et al., 2016).
A decrease in the deep convection intensity since the WMT (Houpert et al., 2016; Li and Tanhua, 2020),
could potentially lead to the reduction in the supply from the nutrient-rich DW (before WMT) to the
surface, i.e. the decrease in nutrient could have happened right after the WMT in spring 2005 where
Schroeder et al. (2010) reported peculiar divergence between the old WMDW and the new WMDW in
nitrate and phosphate; the new WMDW was low in nutrient; later on an intense DW formation event
marked the year 2012 with a strong ventilation that has been recorded in the Adriatic Sea that could
have affected the WMED. It was not possible to observe this change since we calculated the mean state
of the basin spanning specific period.
The spatial distribution of nutrient concentrations after the WMT (2005-2017) was quite different from
the one before the WMT (1981-2004). This could also be related to the significant decline in river
discharge between 1960 and 2000, that was estimated to 20% (Ludwig et al., 2009). The change could
be explained by the low denitrification rate for nitrate and an increase in the remineralization of organic
matter, loading the deep layer with inorganic nutrients, also it could be associated with the slower
ventilation of the WMED waters and a longer residence time.

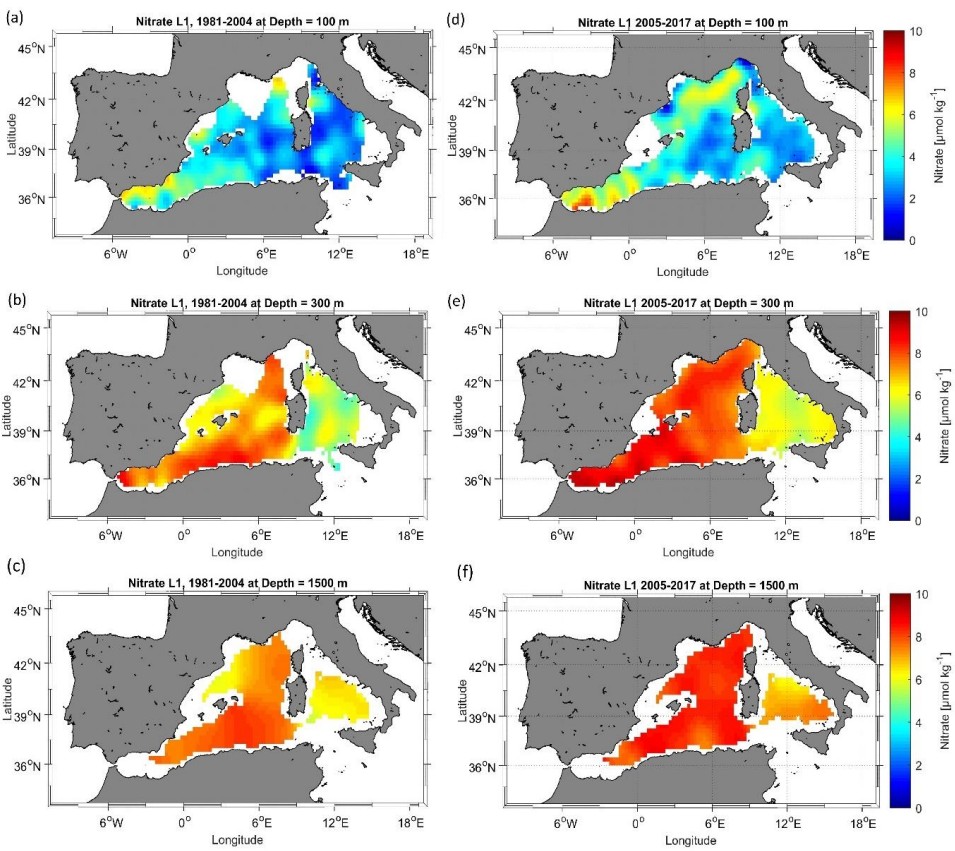


**Figure 18**. Nitrate climatological field (masked analysis fields masked using relative error threshold =

0.3 (L1)) at 100 m, 300 m, and 1500 m, for two periods: 1981-2004 (a, b, c) and 2005-2017 (d, e, f).



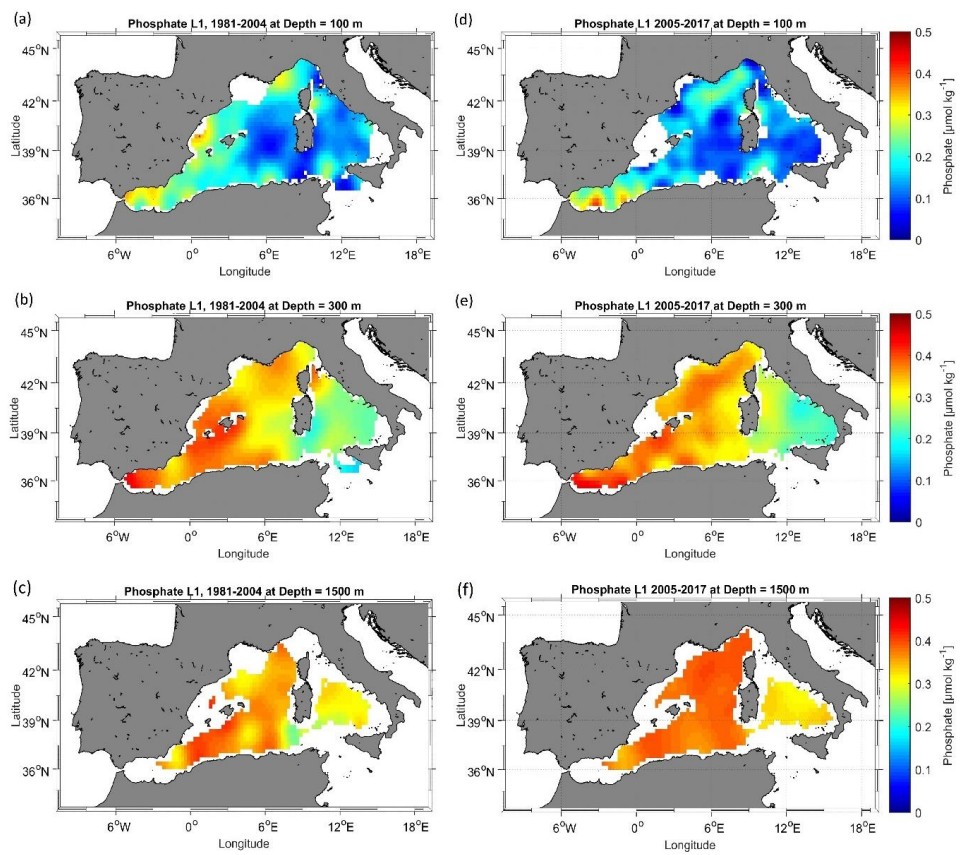


**Figure 19.**The same as Fig. 18 but for phosphate.

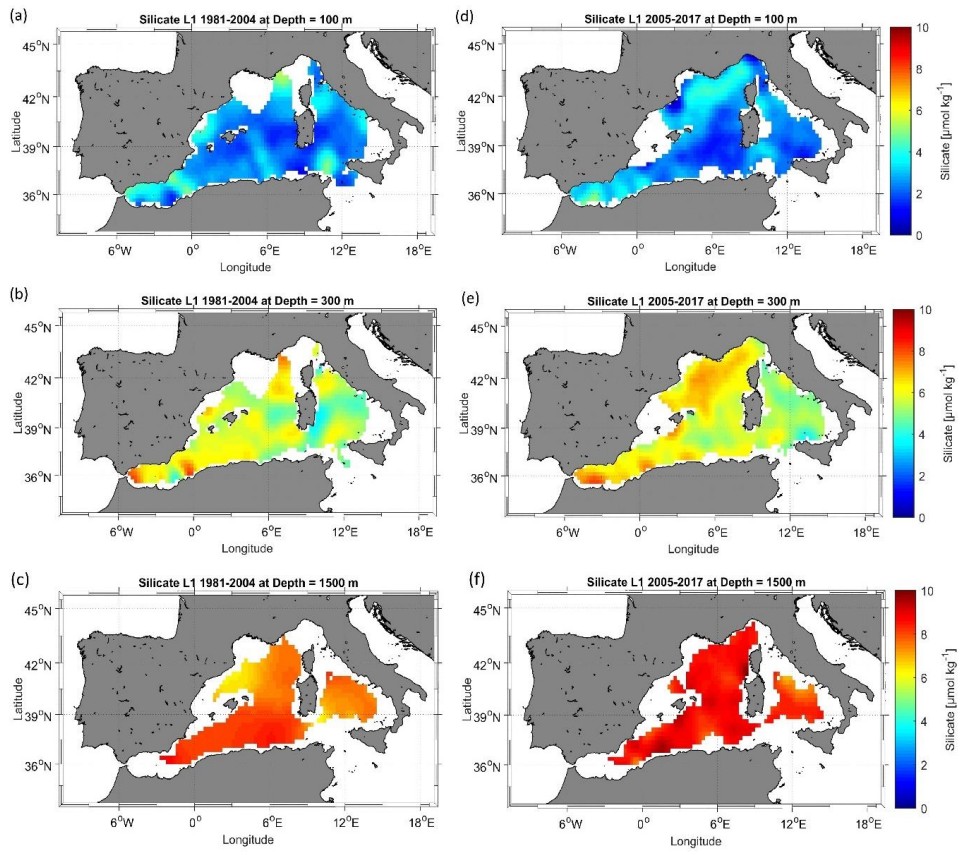


**Figure 20**. The same as Fig. 18 but for silicate.

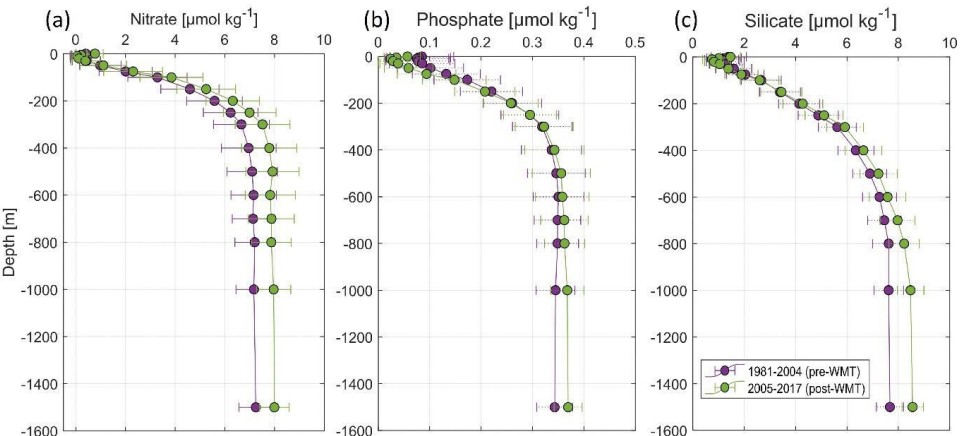


**Figure 21**. Climatological mean vertical profile and standard deviation of (a) nitrate, (b) phosphate and

(c) silicate over the WMED before (1981-2004, in violet) and after WMT (2005-2017, in green).





**5 Data availability**

The final product is available as netCDF files from PANGAEA and can be accessed at https://doi.pangaea.de/10.1594/PANGAEA.930447 (Belgacem et al., 2021, DOI registration in progress). Ancillary information is in the readme in PANGAEA with the list of variables that is described in table 3 of section 4.

**6 Conclusion**

In this study, we investigated spatial variability of the inorganic nutrients in the WMED and present a climatological field reconstruction of nitrate, phosphate, and silicate, using an important collection dataset spanning 1981 and 2017. The BGC-WMED new product is generated on 19 vertical levels on 1/4° spatial resolution grid.

The new product represents very well spatial patterns about nutrient distribution because of its higher spatial and temporal data coverage compared to the existing climatological products (see Table 1), it is contributing to the understanding of the spatial variability of nutrients in the WMED.

The novelty of the present work is the use of the variational analysis that takes into consideration physical, geographical boundaries and topography, the resulting estimate of associated error field.

Comparison with previously reported studies gives that the BGC-WMED reproduces common features and agrees with previous records. The reference products WOA18 and medBFM biogeochemical reanalysis tend to underestimate nutrient distribution in the region with respect to the new product.

The new product captures the strong east-west gradient of and vertical features. The results obtained do not include seasonal or annual analysis fields. However, the aggregated dataset here does show improvements in describing the spatial distribution of inorganic nutrients in the WMED. We acknowledge that computing a climatological mean over a time period is not enough to estimate and detect the climate shift 'WMT' change driven trend. However, comparing climatologies based on the two time periods: 1981-2004 (pre-WMT) and 2005 -2017 (post-WMT) has already produced important results. Notable changes have been found in nutrient distribution after the WMT at various depths.

The results support the tendency to a relative increasing load of inorganic nutrients to the WMED and possibly relate the change in general circulation patterns, changes in deep stratification and warming trends, however, this remains to be evidenced.

The BGC-WMED is a regional climatology that has allowed the identification of a substantial enrichment of the waters, except for the Tyrrhenian Sea where the water column is depleted in nutrients with respect to the western areas of the WMED. The climatology gave information about the spreading of inorganic nutrients inside the WMED at surface, intermediate and deep layers.



A future work will suggest a better understanding of the change in nutrients related to water masses
associated with ventilation rate, a climatological field along isopycnal surfaces instead of depths and the
correlation between potential temperature and nutrients.
**Author contributions**
The BGC-WMED climatology product was led between the CNR-ISMAR and DAIS- University of
Venice. MB, KS and JC designed the experiment and contributed to the writing of the manuscript. AB
and CT helped MB to perform the analysis and contributed to the manuscript. BP contributed to
specific parts of the manuscript.
**Acknowledgements**
Data was provided through SeaDataNet Pan-European infrastructure for ocean and marine data
management (https://www.seadatanet.org), Mediterranean Ocean Observing System for the
Environment, MOOSE (http://www.moose-network.fr/). MB acknowledge the WOA18 and CMEMS
for the medBFM data (https://help-cmems.mercator-ocean.fr/en/articles/4444611-how-to-cite-or-
reference-copernicus-marine-products-and-services ). We wish to thank all colleagues who contributed
in the data acquisition, and the PIs if the cruises involved. MB thanks Kanwal Shahzadi from the
university of Bologna for the discussions during our internship at GHER, university of Liege. JC and
KS acknowledge several of national and European projects, e.g.: KM3NeT, EU GA #011937; SESAME,
EU GA #GOCE-036949; PERSEUS, EU GA #287600; OCEAN-CERTAIN, EU GA #603773;
COMMON SENSE, EU GA #228344; EUROFLEETS, EU GA #228344; EUROFLEETS2, EU GA #
312762; JERICO, EU GA #262584; the Italian PRIN 2007 program "Tyrrhenian Seamounts
ecosystems", and the Italian RITMARE Flagship Project, both funded by the Italian Ministry of
University and Research.

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

**Appendix**

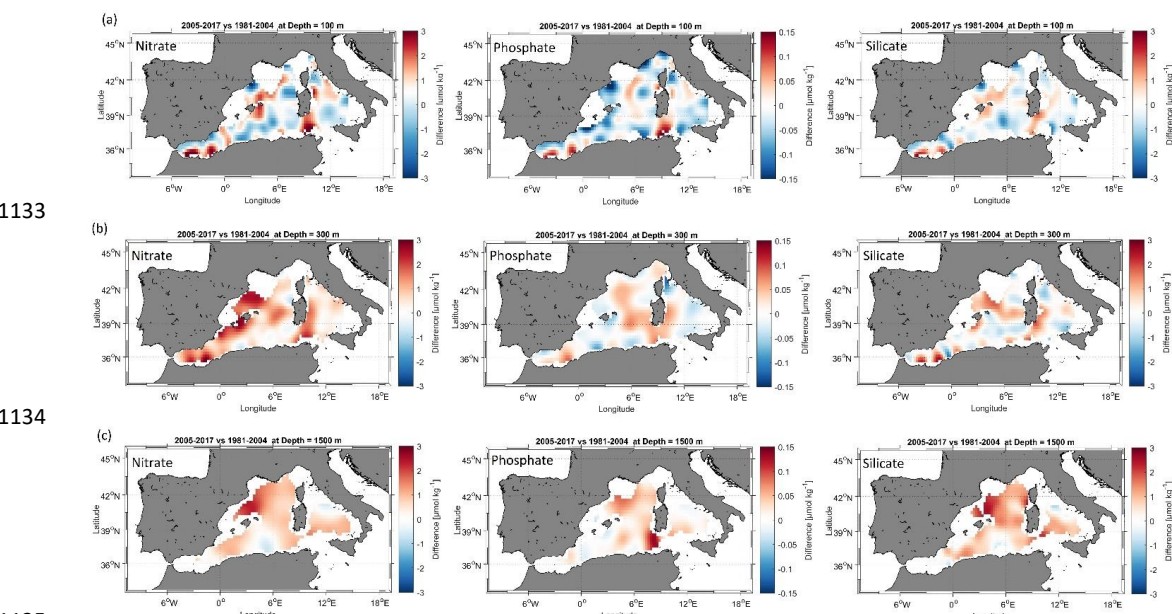



**Figure A1**. (a) Difference field at 100 m between the 1981-2004 climatology and the 2005-2017
climatologies; (b) Difference field at 300 m (c) Difference field at 1500.