# Peer review of "Climatological distribution of dissolved inorganic nutrients"

_Earth System Science Data, 2021_

## Author Comment (AC4)

On behalf of all authors, we would like to thank the reviewer for the comments on the value of the new product and on the proposed suggestions on the manuscript.

Below the reviewer's comments are given in black font and responses in blue font.

**Referee #1 :**

**General comments**

The article shows the exhaustive work of data validation and quality check performed before the analysis and climatological fields computation. Explanations are easy to follow, and the entire work could be replicated by other people in other marine areas. The analysis of results brings out the main differences and agreements between this new computed climatologies and the existing ones (WOA2018, medBFM) with carefully explained details level by level and basin by basin.

The lack of data in certain areas is a well-known problem in the Mediterranean and researchers have to face it when they try to find complete descriptions or intermediate and deep sea layers behaviors. The problem is greater when they want to obtain information about nutrients at intermediate or deep water measurements. In this sense, these climatologies could help in future works to partially palliate this lack of data by offering reference values and error fields. Future updates an re-evaluation of these climatologies will be welcomed by the scientific community.

**Specific comments**

Regarding the GEBCO-30sec (aprox 920mx920m grid) dataset used by the authors, I would put forward that the current EMODnet Digital Terrain Model (DTM), released in Sept2018 has a grid resolution of 1/16 minute x 1/16 minute (=115m x 115 m) and new DTMs are planned to be published in short. As GEBCO, it is freely available to any interested people. Note also that from the start of the EMODnet Bathymetry projects, the followed approach is based in the pre-gridded DTMs. They have been locally computed by the data providers according to the chosen grid and origin. This facilitates the precision of the final computed DTM. This bathymetry, perhaps, could improve the presented climatologies at some levels and can be is a line to explore in future works.

Of course, we thank the reviewer for this suggestion. The EMODnet Digital Terrain could be integrated. Though, even if EMODnet has a better resolution, it is not needed here: the bathymetry will be re-computed on the same grid as the climatology, so it is better to have already a bathymetry with a resolution close to the target grid size. In other regions, we have tested different bathymetry resolutions, the best results were obtained when bathymetry resolution is close to climatology resolution.

In relation with the Deep Water formation in the Gulf of Lion, the importance of Tester's team and work (line53) must be recognised, and the great impulse to the knowledge of this phenomenon achieved with the new instrumental advances, but I would like to draw the attention of the authors to the extensive bibliography published along the time, to which they could also refer. I mention some articles as an example below, but they are not the only ones.

- MEDOC Group : Observation of formation of Deep Water in the Mediterranean Sea. Nature, 227, pp. 1037-1040, 1970.
- Canals, P. Puig, X.D. de Madron, S. Heussner, A. Palanques, J. Fabres. Flushing submarine canyons Nature, 444, pp. 354-357, 10.1038/nature05271, 2006.
- Durrieu de Madron, X., L. Houpert, P. Puig, A. Sanchez-Vidal, P. Testor, A. Bosse, C. Estournel, S. Somot, F. Bourrin, M.N. Bouin, M. Beauverger, L. Beguery, A. Calafat, M. Canals, C. Cassou, L. Coppola, D. Dausse, F. D'Ortenzio, J. Font, S. Heussner, S. Kunesch, D. Lefevre, H. Le Goff, J. Martín, L. Mortier, A. Palanques, P. Raimbault. Interaction of dense shelf water cascading and open-sea convection in the northwestern Mediterranean during winter 2012: Shelf Cascading and open-sea convection. Geophys. Res. Lett., 40, pp. 1379-1385, 10.1002/grl.50331, 2013.

We agree, we added more references about the deep water formation in the northwestern Mediterranean in the revised version.

In the QC of the analysis fields (line 339) the authors refer that the residual values are NaN. This could be a computational output, but it could be explained in another way. It will be preferable if they explain that these computational points are not considered by different reasons (out of domain, or whatever). In my humble understanding, NaN is merely the way in which the computational tool used (julia) encodes the criteria adopted by the authors. It will be very welcomed if the authors consider to reformulate the paragraph.

Yes, the paragraph was reformulated in the revised version.

**Technical corrections**

line 226. Please add the link / reference to Diva User Guide

Done.

line 697. Please type Western Mediterranean Transition (WMT)

Done.

line 779. Please, add the corresponding DOI to reference

Done.

**Referee #2 :**

This is a well-written, high-quality manuscript providing a significant long-term climatological dataset for the biogeochemical parameters, their spatio-temporal change and the processes involved in the Western Mediterranean. The data product is based on in-situ observations from various cruises over the period 1981 – 2017.

On behalf of all authors, I would like to thank the reviewer for the comments on the value of the new product and on the proposed suggestions on the manuscript.

1. The abstract is informative, although I would like to see 1-2 statements on data quality, on the comparison with previous datasets and the overall value of this data product.

Done.

2. In Introduction, lines 31-32, I believe upwelling is not relevant in the context of this sentence.

We adjusted the text accordingly.

3. In Introduction, the main physicochemical processes and circulation affecting the distribution of nutrients in the WMED is shown.

   However, the role of rivers supplying nutrients in the EMED and WMED is not discussed. Please add a paragraph discussing the main riverine fluxes of nutrients, mostly from the Nile, Rhone, Po and Ebro, and their trends in the various basins and sub-basins.

We added more details about rivers in the WMED and how it can affect nutrient variability.

4. Table 1 illustrates the existing two nutrient climatologies for the WMED. The Table is informative and well-structured; it is unclear however, if the present work contains also the data of the past climatologies.

The present work contains some observations from SeaDataNet that were for sure used to generate the EMODnet climatology. Some of these observations are also present in MEDAR/Medatlas, and maybe those observations were submitted to the World Ocean Database (WOD). From the analysis that we designed, we can ensure that we are bringing new data in the BGC-WMED product, not available in WOD.

Please explain. Also, the WOA18 contains only bottle data, while you imply to also use data from argo-floats. It is unclear what are the percentages between bottle data and data from Argos.

We used only CTD profiles and bottle data acquired during cruises which were part of national and international databases. We have added additional details in the text in section 4.3.

The work is a starting point for a living data product that can be followed by another version including more data from argofloats.

Finally, please provide the units used to report the nutrient data per climatology.

We updated table. 1 accordingly.

5.  One deficiency of the database built is the gap in observations between 1997 and 2003. Although mentioned and presented graphically in Figure 2a, it is not very clear how authors deal with this gap in their analysis.

I did not, since the resulting gridded fields were computed over predefined time periods. This gap shaped the experiment. Considering the uneven distribution of observations in time and space (there are spatial and temporal gaps), the climatological gridded fields were computed over three time periods (1981-2017, 1981-2004, 2005-2017).

6.  Another deficiency is the bias towards summer period. Again, it is not very clear how mean-annual fields are produced from these biased data.

Fig.2a shows the monthly distribution of nutrient observations. Measurements were mainly sampled during the warm months, that's why we described the resulting climatology as being more representative of the warm season of the year.

The annual fields are computed by taking all the data, whatever the period, all together.

The statement "Adjustments were applied to measurements when bias was detected" should be further elaborated.

The bias in the season is not the same as the bias discussed in section 3.2 about quality control; details were added in the revised version to clarify the statement.

7. Authors do not report the analytical methods followed for nutrient analysis, per database used. Perhaps a Table including the analytical techniques and instrumentation used will be useful.

The analytical methods and the used instruments do not only change from one dataset to the other but also from cruise to cruise. So, it is difficult to enumerate all of them. In the revised version, I added supplementary materials (Table S1.) where I identify references to the metadata and papers dealing with the instrumentation used with the analytical techniques and QC adopted to set up the dataset. The analytical methods of the BGC-WMED data product can be found in Belgacem et al. (2020).

8. For the data quality check, perhaps a flow chart will guide the readers on the step-by-step procedures followed. In WOA18, other checks are also followed, like the Range and Gradient Check and the Representativeness of the data check. It is unclear if QA/QC followed here is the same as the compared climatologies.

Thank you for the suggestion, which we think is a great addition to the paper. We did not follow the WOA18 checks, but a quality check was first focused on the quality of the observations from the various datasets then another quality check of the resulting climatological fields described in section 3.3 and 3.4. a flowchart is added to the text (Fig. 4).

9. In Eq (1) the term (μi) in the observational constraint term is not explained. What is its range and how it is evaluated. Similarly, although known, it is better to describe the φ(xi, yi)-term.

The first term (observation constraint) considers the distance between the observations and the analysis reconstructed field $\varphi(x_i, y_i)$, so that $\mu_i$ penalizes the analysis misfits relative to the observations. If the observation constraint is only composed of $d_i - \varphi(x_i, y_i)$, the constructed field would be a simple interpolation of the observations and the minimum is reached when $d_i = \varphi(x_i, y_i)$. The field $\varphi(x_i, y_i)$ need to be close to the observation and not have abrupt variation. The observational constraint $\mu_i$ is directly related to the error variance of the observation parameter (see table 1).

"Qualitatively validated the impact of this parameter as a compromise between over-fitting and under-fitting".

This parameter can be optimized by cross-validation, but it is not trivial to take the spatial correlation between observation's error into account. Also, it is quite expensive in terms of CPU time.

The revised version was adapted to explain better the terms in Eq. (1).

10. Authors state that they applied the fourth-dimensional DIVAnd method. What is DIVA's response when there are gaps in time as in this dataset. Perhaps analysis should be divided in two periods, prior and after the period lagging data, and apply DIVA in each separate dataset.

Yes, we did it. Indeed, we propose three climatologies for the whole observational period (1981-2017), and two sub-intervals (1981-2004, 2005-2017).

Note that actually in diva3d, one uses a 3D analysis (lon/lat/depth). If there are large gaps, the analysis would revert to the background estimate.

11. Also, please explain if DIVA was used to extrapolate data in areas with data gaps and towards the coast?

A land-sea mask is created using the coastline and bathymetry. The reconstructed fields are determined at the elements of a grid on each isobath using the cost function Eq. (1). The grid is dependent on the correlation length and the topographic contours. The variational computation introduces a cost function $J[\varphi]$ to penalize the misfit in the resulting gridded field such as the variation from the background estimate field. It is the first guess. In our analysis, we used the default field which consists of spatial data average for each isobath.

Along with the gridded fields, DIVA yields error fields dependent on the data coverage and the noise in the measurements. In Fig.8b, the example of the relative error field is shown, where the error increases (shown in orange-red, Fig.8b) due to the sparsity of the observations in some areas.

Given the inhomogeneous distribution and the sparsity of ocean observation, extrapolating observation is unavoidable if one aims to generate a full grid. Using a first guess (or also called a background estimate) which is updated by nearby observations (is present) guards against unrealistic extrapolations.

12. Line 261. For which parameter do these values refer? for which depth?

We mentioned it in Figure 5, but we added more details in the text.

13. Lines 277 – 279. Probably here you refer to the horizontal Lc variability. How do you comment on the lower values of Lc for silicates compared to nitrate and phosphate?

I think that Lc for silicate has lower values compared to nitrate and phosphate, because, horizontally and vertically, it behaves in a different way. Unlike nitrate and phosphate, silicate does not show a strong east-west increased gradient. This gradient might induce this difference in the horizontal distance over which the sample

influences its neighbourhood (see the scatterplot of the observation at a fixed depth e.g. 600m below). We explained this difference better in the revised version.

*Scatterplot of the observations at 600m:*

[Figure]

14. Lines 280 – 288. Text is mixing the diagrams and Lc-parameter from horizontal to vertical. Better write one paragraph describing the horizontal and another for the vertical Lc.

The paragraph describing the two Lc is added to the revised version.

15. Line 330. "A score is assigned to each observation". Please elaborate on the score assigned per observation. What is the score range and the increments used on the scaled error.

The score and quality assurance are added to the supplementary materials.

16. Figure 6. Please explain the dashed blue line.

We added more explanation in the revised version.

17. Lines 389-391. You should also refer to the rivers supplying nutrients to each area. The e-HYPE database from SMHI could be helpful to assess the riverine fluxes of nutrients.

We are aware that the part related to the rivers is lacking, the proposed reference is very important, we appreciate it. Rivers can be considered as a "forcing" term here, meaning that they are influencing the concentrations near the river mouths, but here the goal is to create fields from the in situ observations, not to model the influence of rivers. We refer to the main rivers in the revised version. In future works, we can investigate and relate the climatological outputs to the riverine fluxes.

Since nitrate is the dominant N-species in the WMED, authors could produce the stoichiometric N:P:Si ratios and discuss their mean and standard deviation values per sub-area and for the whole WMED.

We have added a new product to the well-known existing nutrient climatologies. The main purpose of the paper is to make available climatological products for each nutrient and test its reliability. In our future studies (the reviewer is more than welcome to be part of the proposed research) we aim at exploring more the ratios and producing the stoichiometric N:P:Si and also adding the dissolved oxygen components for a better understanding of the biogeochemical processes in the region.

18. Line 506. Please consider that medBFM assimilates satellite and argo data and includes terrestrial inputs of N and P from 39 rivers.

We have added a sentence in section 4.3 describing better the medBFM reanalysis.

19. Lines 516 – 522. If I understand well, you re-gridded the BGC-WMED from 0.25 deg to 1 deg, to compare with WOA18 and the medBFM from, 0.063 deg to 0.25 deg to compare with BGC-WMED. Please explain better this process.

We have made the necessary changes and explained the process better.

20. Lines 542-543. Could the largest difference seen in the Alboran Sea be attributed to the occurrence of more frequent upwelling events during the WOA18 period?

Yes, but I don't think so, because the higher values of nitrate and phosphate have been recorded in the BGC-WMED (Fig.10a, Fig.11a), so the largest difference seen in the Alboran Sea could be during the period of the new climatology. More upwellings resulted in an important injection of nutrient to the surface layer that the WOA18 did not show it with details, it showed lower concentration in the same subregion. The section has been proofread to explain this difference better.

21. Line 761-762. This is a valid point you are making on the decline of river discharge. What about the nutrient fluxes over this period?

We did not describe much the nutrient fluxes in the region, because the focus was the data product. In the revised version, we added more information related to nutrient fluxes.

**CC#1, CC#2 :**

We thank Laurent Coppola and Simona Simoncelli for highlighting the missed sources. We added the references, and a better description of the MOOSE data in the final version of the paper.

---

## Author Response (AR2)

Thank you.

The editor's the suggestions were implemented.

**Comments to the author**:
The paper is almost ready for publication. I ask you only a couple of adjustments in order to take into account also updates in publications you mentioned.
1) line 67: to the Giorgi 2000 reference, I recommend adding the recent paper Upper Ocean Temperatures Hit Record High in 2020 by Lijing Cheng et al., (2021) https://link.springer.com/article/10.1007/s00376 -021-0447-x

Done .
2) Line 109: GLODAP has been continuously updated and I suggest adding references to new versions
- An updated version of the global interior ocean biogeochemical data product, GLODAPv2.2020, by Are Olsen et al
https://essd.copernicus.org/articles/12/3653/2020/essd-12-3653-2020.pdf
- An updated version of the global interior ocean biogeochemical data product, GLODAPv2.2021 by Siv K. Lauvset et al, https://essd.copernicus.org/preprints/essd-2021-234/
There is no need to update the list of data sources in table A1

The GLODAP references has been updated as well.